# DUALFORMER: DUAL GRAPH TRANSFORMER

**Jiaming Zhuo**[1]  **Yuwei Liu**[1]  **Yintong Lu**[1]  **Ziyi Ma**[1]  **Kun Fu**[1]  **Chuan Wang**[2]
**Yuanfang Guo**[3]  **Zhen Wang**[4]  **Xiaochun Cao**[5]  **Liang Yang**[1]*
[1]Hebei Province Key Laboratory of Big Data Calculation,
  School of Artificial Intelligence, Hebei University of Technology, Tianjin, China
[2]School of Computer Science and Technology, Beijing JiaoTong University, Beijing, China
[3]School of Computer Science and Engineering, Beihang University, Beijing, China
[4]School of Artificial Intelligence, OPtics and ElectroNics (iOPEN),
 School of Cybersecurity, Northwestern Polytechnical University, Xi'an, China
[5]School of Cyber Science and Technology,
  Shenzhen Campus of Sun Yat-sen University, Shenzhen, China
`jiaming.zhuo@outlook.com,`
`{202332803037, 202322802030}@stu.hebut.edu.cn,`
`zyma@hebut.edu.cn, fukun@hebut.edu.cn, wangchuan@iie.ac.cn,`
`andyguo@buaa.edu.cn, w-zhen@nwpu.edu.cn,`
`caoxiaochun@mail.sysu.edu.cn, yangliang@vip.qq.com`

## ABSTRACT

Graph Transformers (GTs), adept at capturing the locality and globality of graphs, have shown promising potential in node classification tasks. Most state-of-the-art GTs succeed through integrating local Graph Neural Networks (GNNs) with their global Self-Attention (SA) modules to enhance structural awareness. Nonetheless, this architecture faces limitations arising from scalability challenges and the trade-off between capturing local and global information. On the one hand, the quadratic complexity associated with the SA modules poses a significant challenge for many GTs, particularly when scaling them to large-scale graphs. Numerous GTs necessitated a compromise, relinquishing certain aspects of their expressivity to garner computational efficiency. On the other hand, GTs face challenges in maintaining detailed local structural information while capturing long-range dependencies. As a result, they typically require significant computational costs to balance the local and global expressivity. To address these limitations, this paper introduces a novel GT architecture, dubbed DUALFormer, featuring a dual-dimensional design of its GNN and SA modules. Leveraging approximation theory from Linearized Transformers and treating the query as the surrogate representation of node features, DUALFormer *efficiently* performs the computationally intensive global SA module on feature dimensions. Furthermore, by such a separation of local and global modules into dual dimensions, DUALFormer achieves a natural balance between local and global expressivity. In theory, DUALFormer can reduce intra-class variance, thereby enhancing the discriminability of node representations. Extensive experiments on eleven real-world datasets demonstrate its effectiveness and efficiency over existing state-of-the-art GTs.

## 1 INTRODUCTION

Node classification, aimed at accurately predicting the categories or labels of individual nodes based on the graph topology and node attributes, is an essential task in social networks Bhagat et al. (2011), citation networks Ju et al. (2024), and biological sciences Yi et al. (2022), among other domains. Graph Neural Networks (GNNs) Kipf & Welling (2016) have emerged as a dominant architecture for this task due to their outstanding ability to capture local neighborhood information. Most GNNs, such as the seminal GCN Kipf & Welling (2016), and subsequent innovations Klicpera et al. (2019); Velickovic et al. (2017) follow a graph-based message-passing paradigm Gilmer et al. (2017), where

---

*corresponding author

the representations of each node are updated by aggregating the information from its neighboring nodes. Although such localizing property has propelled notable success for GNNs, the other side is that it creates a bottleneck for GNNs to obtain long-range dependencies Dwivedi et al. (2022). As a result, GNNs often encounter over-smoothing and over-squashing issues Alon & Yahav (2021).

Initially developed for NLP, Transformers Vaswani et al. (2017), known for full-token connectivity enabled by the Self-Attention (SA) mechanism, have shown significant performance across various tasks Gillioz et al. (2020). Recently, Graph Transformers (GTs) with excellent expressivity have been presented and shown notable potential in addressing the above issues of GNNs Wu et al. (2021); Chen et al. (2022; 2024a;b). The success of GTs lies in their ability to simultaneously capture long-range dependencies and structural bias from graph structure. Accordingly, GTs for node-level tasks can be broadly divided into four categories: 1) Node attribute extension using positional/structural encoding, including NodeFormer Wu et al. (2022), NAGphormer Chen et al. (2023), Exphormer Shirzad et al. (2023), and GOAT Kong et al. (2023); 2) Sampling context nodes based on graph, containing GOAT and NAGphormer; 3) GNN block integration, encompassing NAGphormer, GOAT, SGFormer Wu et al. (2024), Polynormer Deng et al. (2024), and CoBFormer Xing et al. (2024); and 4) Edge rewriting, represented by Exphormer. Their introduction signifies an initial stride.

Nonetheless, existing GTs typically encounter two primary challenges, as illustrated in Fig. 1(a). *1) Unscalability.* In pursuit of global dependencies between nodes, GTs, particularly those built upon the vanilla SA mechanism Wu et al. (2021); Chen et al. (2022), tend to encounter scalability issues due to their quadratic time and space complexity, which is prohibitive when dealing with large-scale graphs. To address this issue, existing approaches tend to compromise the global expressivity (e.g., NAGphormer and Exphormer) or increase model complexities (e.g., GOAT, CoBFormer). This results in models that may fail to generalize well. *2) Tradeoff dilemma of locality and globality.* It is noteworthy that several state-of-the-art GTs (*e.g.*, NAGphormer, GOAT, SGFormer, Polynormer, and CoBFormer), either explicitly or implicitly, still resort to the GNNs to learn local node representations; These representations are then incorporated with Self-Attention (SA) blocks to generate final node representations. Unfortunately, GTs demonstrate a certain level of information loss, stemming from the necessity to trade off the locality and globality. Specifically, they tend to encounter challenges in retaining the fine-grained structural details while capturing long-range dependencies, which could lead to the loss of nuanced information crucial for accurate node classification.

This paper seeks to break the aforementioned limitations by devising a novel GT architecture with (1) an efficient SA mechanism and (2) a comprehensive fusion mechanism of local and global information. To this end, the architectures of existing GTs are systemically investigated, and an intuitive interpretation of these limitations from a decoupled perspective, as shown in Fig. 1(a). Specifically, the existing global SA mechanisms default to the value as the agent representation of node features, essentially performing a global message passing between nodes to capture the global dependencies. Thus, the common fundamental factor in these two issues is an over-reliance on a single dimension. Intuitively, through regarding the query as the agent representation and leveraging the established approximation between (Query×Key)×Value and Query×(Key×Value), the aforementioned global message passing can be efficiently executed along the feature dimension. Inspired by these insights, this paper proposes DUALFormer, a simple yet effective GT architecture featuring a dual-dimension design, as depicted in Fig. 1(b). The idea of DUALFormer is to decouple the GNN and SA modules from the same dimension to separately model local and global information in the node dimension and feature dimension, respectively. Therefore, it achieves efficient global SA on the one hand, and on the other, it naturally integrates locality and globality without compromising the trade-off. The comparisons between GTs regarding performance, running time, and GPU memory usage demonstrate the effectiveness and efficiency of DUALFormer, as shown in Fig. 2.

The contributions of this paper can be summarized as follows:

- We investigate two drawbacks of existing Graph Transformers (GTs), that is, unscalability and tradeoff dilemma between local and global expressivity.

- We introduce DUALFormer, a simple yet comprehensive GT architecture featuring a dual-dimensional design of its local and global modules.

- Extensive experiments on public benchmark datasets demonstrate the superior performance and efficiency of DUALFormer over state-of-the-art GNNs and GTs.

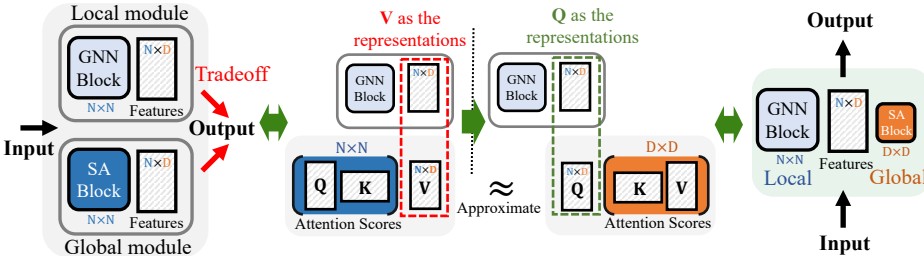

(a) Existing GTs with a tradeoff strategy for local and global modules.

(b) The proposed DUALFormer with a simple yet comprehensive dual-dimensional architecture.

Figure 1: The design motivation of the proposed DUALFormer and its comparison with existing state-of-the-art GT architectures. (a) Existing GTs suffer from two primary challenges: 1) the scalability issue from Self-Attention (SA) mechanisms, and 2) the tradeoff dilemma of local and global information. By default, global SA mechanisms serve the value $\mathbf{V}$ as the agent representations of node features, so that by employing attention score matrix $sim(\mathbf{Q}, \mathbf{K}) \in \mathbb{R}^{n \times n}$ to capture global dependencies among nodes. Intuitively, by leveraging the approximation $sim(\mathbf{Q}, \mathbf{K})\mathbf{V} \approx \mathbf{Q}sim(\mathbf{K}, \mathbf{V})$, and treating the query $\mathbf{Q}$ as the agent representations, the above global SA mechanism can be efficiently implemented in the feature dimension. (b) DUALFormer is a dual-dimensional GT architecture that seamlessly integrates the local GNN block and global SA block on dual dimensions. Thus, DUALFormer effectively and comprehensively leverages the advantages of both dimensions.

## 2 PRELIMINARIES

This section starts with an overview of the notation used in this paper. It then provides an introduction to the fundamental concepts of Graph Neural Networks (GNNs). Finally, it details the vanilla Transformer architecture and Graph Transformers (GTs) that extends upon GNNs.

### 2.1 NOTATIONS

Matrices (e.g., $\mathbf{W}$) are in bold capital letters, vectors (e.g., $\mathbf{w}_{i,:}$, which represents the $i$-th row vector of $\mathbf{W}$), are in bold lowercase letters, scalars (e.g., $w_{i,j}$, which stands for the elements in the $i$-th row and $j$-th column of $\mathbf{W}$), are in lowercase letters, and sets (e.g., $\mathcal{V}$) are in calligraphic letters.

For a general-purpose description, this paper examines an undirected attribute graph $\mathcal{G}(\mathcal{V}, \mathcal{E})$. The node set $\mathcal{V}$ consists of $n$ node instances $\{(\mathbf{x}_v, \mathbf{y}_v)\}_{v \in \mathcal{V}}$, where $\mathbf{X} \in \mathbb{R}^{n \times f}$ and $\mathbf{Y} \in \mathbb{R}^{n \times c}$ represent the attributes and labels of node $v$, respectively, and $f$ is the attribute dimension and $c$ is the label dimension. $\mathcal{E} = \{(v_i, v_j)\}$ stands for the edge set. Generally, the adjacency matrix $\mathbf{A} \in \mathbb{R}^{n \times n}$ is utilized to describe the graph topology and $a_{i,j} = 1$ only if there is the edge $(v_i, v_j) \in \mathcal{E}$, otherwise $a_{i,j} = 0$. Its forms of Random-Walk (RW) normalization $\hat{\mathbf{A}} = \mathbf{D}^{-1}\mathbf{A}$ and Laplacian normalization $\hat{\mathbf{A}} = \mathbf{D}^{-\frac{1}{2}}\mathbf{A}\mathbf{D}^{-\frac{1}{2}}$ are commonly utilized, where $\mathbf{D}$ denotes the diagonal degree matrix. Moreover, the graph $\mathcal{G}$ can be expressed in matrix form as $\mathcal{G}(\mathbf{A}, \mathbf{X})$. Besides, $\mathbf{Y}_{\mathcal{L}} \in \mathbb{R}^{n_u \times c}$ and $\mathbf{Y}_{\mathcal{U}} \in \mathbb{R}^{n_u \times c}$ stand for the labels of the labeled and unlabeled nodes, respectively.

### 2.2 GRAPH NEURAL NETWORKS

Propagation-based Graph Neural Networks (GNNs) typically follow the message-passing paradigm Gilmer et al. (2017) to harness structural biases in graphs. Within each layer, the representations of center nodes are updated by iteratively aggregating and combining the features from their adjacent nodes (*i.e.*, one-hop neighbors) in the graph. In specific, for node $v$ in the $l$-th layer, denoted by $\mathbf{h}_v^l$, the above representation update can be expressed as

$$\hat{\mathbf{h}}_v^l \triangleq \text{Aggregation}^l(\{\mathbf{h}_u^{l-1} | u \in \mathcal{N}_v^G\}), \quad \mathbf{h}_v^l \triangleq \text{Combination}^l(\mathbf{h}_v^{l-1}, \hat{\mathbf{h}}_v^l), \quad (1)$$

where $\text{Aggregation}(\cdot)$ and $\text{Combination}(,)$ denote the aggregation and combination modules, respectively. $\mathcal{N}_v^G$ represents the neighbor set of node $v$. Typical GNNs are detailed in Section B.

Despite their localizing property, GNNs still struggle to effectively capture long-range dependencies Dwivedi et al. (2022). A trivial approach is to broaden the receptive field of each node by stacking

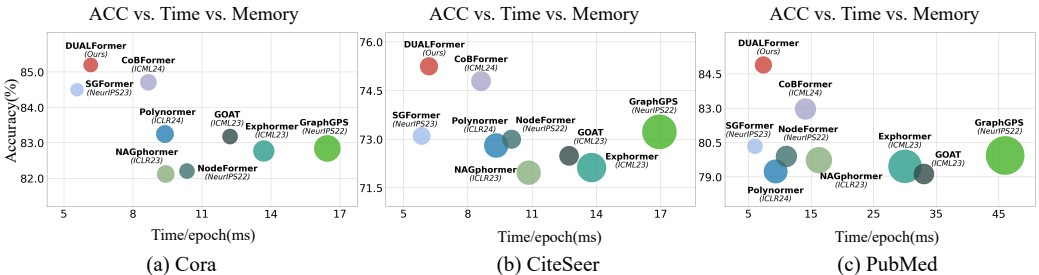

Figure 2: Comparisons between GTs regarding accuracy, running time, and GPU memory usage. The size of circles signifies memory consumption. DUALFormer achieves superior performances in a shorter runtime (except simple SGFormer) over baselines, showing its effectiveness and efficiency.

multiple layers. However, this approach tends to lose discriminative information of representations due to over-smoothing and over-squashing Alon & Yahav (2021).

## 2.3 GRAPH TRANSFORMERS

Unlike the localized GNNs that restrict information aggregation to the neighborhood scope, Transformers facilitate a holistic aggregation across all token pairs through the self-attention mechanism. Intuitively, the above self-attention mechanism functions similarly to GNNs and can specifically be considered as instances with the global receptive field of the message-passing paradigm.

**Transformer Architecture.** The vanilla Transformer Vaswani et al. (2017) consists of two essential components: Multi-Heads Self-Attention (MHSA) and Feed-Forward Network (FFN). The former enables the model to capture global dependencies between all tokens within the sequence, while the latter performs additional non-linear transformation. Given $n$ tokens $\mathbf{Z} = [\mathbf{z}_i]_{i=0}^{n-1} \in \mathbb{R}^{n \times d}$ at each head, the self-attention first maps the input features $\mathbf{Z}$ to query ($\mathbf{Q}$), key ($\mathbf{K}$), and value ($\mathbf{V}$) vectors. Then, attention scores from query-key pairs are applied to weight aggregate the value vectors.

Specifically, a general expression of Self-Attention (SA) module can be expressed as

$$\mathbf{Q} = \mathbf{Z}\mathbf{W}_Q, \ \mathbf{K} = \mathbf{Z}\mathbf{W}_K, \mathbf{V} = \mathbf{Z}\mathbf{W}_V, \tag{2}$$

$$\hat{\mathbf{z}}_v \triangleq \sum_{u \in \mathcal{V}} \mathbf{S}_{v,u}^{QK} \mathbf{v}_u = \sum_{u \in \mathcal{V}} \frac{\exp(sim(\mathbf{q}_v, \mathbf{k}_u))}{\sum_{u \in \mathcal{V}} \exp(sim(\mathbf{q}_v, \mathbf{k}_u))} \mathbf{v}_u, \tag{3}$$

where $\mathbf{W}_Q$, $\mathbf{W}_K$, and $\mathbf{W}_V \in \mathbb{R}^{d \times d}$ denotes trainable projection matrices. $sim(,)$ terms the similarity function, which mainly adopts scaled dot-product attention, *i.e.*, $sim(\mathbf{Q}, \mathbf{K}) = \mathbf{Q}\mathbf{K}^\top / \sqrt{d}$, where $d$ is the feature dimensions. Note that the attention score matrix $\mathbf{S}^{QK} \in \mathbb{R}^{n \times n}$ is derived by computing similarities of all query-key pairs, which leads to the computation complexity of $O(n^2)$.

After obtaining the features for each head ($\hat{\mathbf{Z}}_{(i)}$), MHSA derives the final features by concatenating the outputs from all $t$ heads and then applying a subsequent linear transformation, namely,

$$\hat{\mathbf{Z}}_{final} = \text{Concat}(\hat{\mathbf{Z}}_{(0)}, \hat{\mathbf{Z}}_{(1)}, \ldots, \hat{\mathbf{Z}}_{(t-1)})\mathbf{W}^O, \tag{4}$$

where $\mathbf{W}^O \in \mathbb{R}^{td \times d}$ stands for a trainable projection matrix.

**Transformers on Graphs.** To exploit the structural biases in graphs, current GTs have adopted four strategies to integrate discriminative topology information into the vanilla Transformer, as outlined in Section 1. Among these, many SOTA GTs rely on the GNNs to create local representations, such as SGFormer Wu et al. (2024), Polynormer Deng et al. (2024), and CoBFormer Xing et al. (2024). The final node representations are typically derived via two schemes: ① local-and-global fusion and ② local-to-global fusion, as detailed in Section B.

## 3 METHODOLOGY

This section first highlights the significance of deploying self-attention in the feature dimension by examining its linear approximation. Then, it introduces a dual-dimensional GT architecture. Finally, it compares the proposed GT with existing GTs in terms of scalability, simplicity, and expressivity.

### 3.1 Analysis and Motivations

As previously discussed in the Introduction, existing GTs tend to encounter the scalability issue and the dilemma of trading off global and local information. To overcome the scalability issue, which arises from the need to compute pairwise similarities between queries and keys in Softmax attention, linearized attention mechanisms Katharopoulos et al. (2020); Han et al. (2023) have been introduced. These mechanisms approximate or replace the Softmax attention using separate kernel functions, allowing for changing computation order from the standard (Query×Key)×Value, as formulated by Eq. 3, to a more efficient Query×(Key×Value) format.

Given a kernel function with representations $\phi(X)$, Eq. 3 can be rewritten as

$$\hat{\mathbf{Z}}_v = \frac{\phi(\mathbf{q}_v)\left(\sum_{u\in\mathcal{V}}\phi(\mathbf{k}_u)^\top \mathbf{v}_u\right)}{\phi(\mathbf{q}_v)\left(\sum_{u\in\mathcal{V}}\phi(\mathbf{k}_u)^\top\right)} \tag{5}$$

This equation becomes clearer when expressing the numerator in vectorized form, that is

$$\hat{\mathbf{Z}} = \text{Sim}(\mathbf{Q}, \mathbf{K})\mathbf{V} = \phi(\mathbf{Q})\left(\phi(\mathbf{K})^\top \mathbf{V}\right) \tag{6}$$

where the kernel function can be exponential linear unit $\phi(X) = \text{elu}(X) + 1$. Notably, this mechanism reduces the computation complexity from $O(n^2)$ to $O(n)$.

Upon reviewing the above two forms of self-attention mechanisms, two pivotal insights emerge.

- **The standard self-attention mechanism (Eq. 3) encapsulates the global dependencies among samples.** The formula leverages the value ($\mathbf{V}$) as a surrogate representation of the input features $\mathbf{Z}$, with the attention score matrix $sim(\mathbf{Q}, \mathbf{K})$ functioning as the dependencies matrix between samples. Thus, this formula can be interpreted as a message-passing process that propagates the value features ($\mathbf{v}_{i,:}$) on the global dependencies matrix. Meanwhile, this mechanism possesses global expressivity, *i.e.*, capturing global dependencies.

- **The above linearized self-attention mechanism (Eq. 6) effectively captures the global correlations between features.** From the message-passing viewpoint, one can regard the query ($\phi(\mathbf{Q})$) as the representation of $\mathbf{Z}$, and the product matrix ($\phi(\mathbf{K})\mathbf{V}$) as the correlation matrix between features. In contrast to the original attention (Eq. 3), the linearized attention can be interpreted as another message-passing process between features, which propagates the query features ($\phi(\mathbf{q}_{:,j})$) with the correlation matrix as the propagation matrix. As a result, this mechanism can model the correlations between features.

According to the approximation between these two formulas, it is natural to conclude that *within the self-attention mechanism, characterizing the global dependencies between nodes is approximately equivalent to describing the global correlations between features.* Thus, the self-attention mechanism can be implemented in the feature dimension.

### 3.2 DUALFormer

Inspired by the analysis in the previous subsection, this subsection presents a novel GT transformer, which is referred to as DUAL-dimensional TransFormer (DUALFormer). Central to the design of DUALFormer is an efficient global attention module that captures the implicit dependencies between nodes from the dimension regarding features. DUALFormer includes three modules, each described below. Refer to Section A for the pseudocode of these modules.

**Input Projection Layer.** Considering that the node attributes $\mathbf{X}$ in graph $\mathcal{G}(\mathbf{A}, \mathbf{X})$ are unprocessed and high-dimensional, a Feed-Forward Network (FFN) $f : \mathbb{R}^{n\times f} \to \mathbb{R}^{n\times d}$ is deployed to project these attributes into a low-dimensional hidden space, thus producing informative node features. By opting for Multi-Layer Perceptron (MLP) Rumelhart et al. (1986) as the network architecture, these node features can be represented as

$$\mathbf{H}^0 = \text{MLP}(\mathbf{X}). \tag{7}$$

The obtained feature $\mathbf{H}^0$ serves as the input to the subsequent SA and GNN modules.

To efficiently capture global dependencies between nodes while sidestepping the trade-off dilemma between local and global expressivity, DUALFormer introduces a dual-dimensional architecture, as

shown in Fig. 1. Unlike existing GTs, which typically integrate local and global interactions between nodes only in the node dimension, DUALFormer designs an innovative approach by modeling these interactions across two distinct dimensions. The details are as follows.

**Global Attention Module.** This attention module is designed to efficiently capture global dependencies between nodes. It accomplishes this by introducing a computation unit that captures full-pair correlations among features, thereby enabling the implicit capture of global node interactions. Moreover, the module leverages a comprehensive set of established techniques from the Transformers for each layer, which include the Query-Key-Value computations (Eq. 2), the residual connections He et al. (2016), and the scaled dot-product attention (Eq. 3).

Given the initial node features $\hat{\mathbf{Z}}^{(0)} = \mathbf{H}^0$, this module can be expressed as

$$
\begin{aligned}
\mathbf{Q}^{(l)} &= \hat{\mathbf{Z}}^{(l-1)}\mathbf{W}_Q^{(l)}, \mathbf{K}^{(l)} = \hat{\mathbf{Z}}^{(l-1)}\mathbf{W}_K^{(l)}, \mathbf{V}^{(l)} = \hat{\mathbf{Z}}^{(l-1)}\mathbf{W}_V^{(l)}, \\
\tilde{\mathbf{Z}}^{(l)} &= \mathbf{V}^{(l)}\mathbf{M}^{(l)} = \mathbf{V}^{(l)}\sigma\left(\frac{(\mathbf{Q}^{(l)})^\top \mathbf{K}^{(l)}}{\sqrt{n}}\right), \\
\hat{\mathbf{Z}}^{(l)} &= \alpha\tilde{\mathbf{Z}}^{(l)} \; + \; (1-\alpha)\hat{\mathbf{Z}}^{(l-1)},
\end{aligned}
\tag{8}
$$

where $\mathbf{Q}$, $\mathbf{K}$, and $\mathbf{V}$ denote the query, key, and value, respectively, and $\sigma(\cdot)$ represents the nonlinear activation functions, such as $\mathrm{softmax}(\cdot)$, and $\mathbf{M}$ terms the attention score matrix, which characterizes the feature-to-feature correlations. $\alpha$ denotes a hyper-parameter to balance the representations derived from attention and those from the previous layers. Moreover, to augment the representation capabilities of DUALFormer, this module can incorporate the Multi-Head Self-Attention (in Eq. 4) and layer normalization Lei Ba et al. (2016). For clarity, $\mathbf{V}$ is still used as the agent representation, with the order modified accordingly.

**Local Graph Convolution Module.** After the global representations from the attention module are obtained, diverse graph convolution blocks from GNNs can be incorporated to further integrate local information into the learned representations. Thus, an expression for this module is

$$
\hat{\mathbf{H}}^{(k)} \triangleq \mathrm{GNN}\left(\mathbf{A}, \hat{\mathbf{H}}^{(k-1)}\right),
\tag{9}
$$

where $\hat{\mathbf{H}}^{(0)} = \hat{\mathbf{Z}}^{(L)}$, and $L$ represents the number of the attention layers. $\mathrm{GNN}(,)$ denotes the GNN layers. In DUALFormer, SGC (Eq. 11) is opted for $\mathrm{GNN}(,)$, capitalizing on its streamlined design and effectiveness. Furthermore, the impact of various GNN layers is analyzed in Section 4.2. Once the node representations from $K$ layers of graph convolutions are obtained, a prediction layer is used to yield the final predictions, that is, $\hat{\mathbf{Y}} = \mathrm{MLP}(\hat{\mathbf{H}}^{(K)})$.

The proposed DUALFormer, although simple in design, possesses three attractive characteristics:

- **Efficiency and scalability.** The above architecture, leveraging the sparse diffusion matrix (e.g., adjacency matrices) and low-dimensional attention score matrix $\mathbf{M} \in \mathbb{R}^{d \times d}$, operates with a complexity that is linearly related to the scale of the graph. Therefore, DUALFormer is computationally efficient and potentially scalable to large-scale graphs.

- **Comprehensiveness.** The above architecture thoroughly accounts for the local inter-node dependencies and the global inter-feature correlations, which can be regarded as an approximation of the global inter-node dependencies. Furthermore, owing to the dual-dimensional design, it achieves a natural tradeoff between local and global expressivity. The detailed explanations of the localizing and globalizing properties of DUALFormer are presented in Section E.4. Thus, DUALFormer is an informative and comprehensive GT architecture.

**Theorem 1.** *(Discriminability Improvement.)* *Global attention module (Eq. 8) reduces the intraclass variance while keeping inter-class variance unchanged.*

Theorem 1 indicates that DUALFormer is capable of diminishing class overlap, thus improving the effectiveness of representation learning. The proof of this theorem can be found in Section C.

To sum up, this innovative dual-dimensional design enables DUALFormer to efficiently and comprehensively harness the strengths of both dimensions of the feature matrix within a unified framework. This emphasizes the simplicity and efficiency of DUALFormer.

Table 1: Comparison between GTs in terms of three aspects: Time Complexity, Required Components, including Positional Encodings (**PE**), Augmented Training Loss (**ATL**), Edge Features (**EF**), and Additional Parameters (**AP**), and Global Expressivity. Here, $n$ represents the size of the graphs. $e$ stands for the number of edges. The notation #Appr. represents approximate.

| Model | Time Complexity | | Components | | | | Global Expressivity |
|---|---|---|---|---|---|---|---|
| | Pre-processing | Training | PE | ATL | EF | AP | |
| GraphTrans (*NeurIPS 2021*) | - | $O(n^2 + e)$ | - | - | - | - | YES |
| SAT (*ICML 2022*) | $O(n^3)$ | $O(n^2 + e)$ | ✓ | - | - | - | YES |
| GraphGPS (*NeurIPS 2022*) | $O(n^3)$ | $O(n + e)$ | ✓ | - | ✓ | ✓ | YES |
| NodeFormer (*NeurIPS 2022*) | - | $O(n + e)$ | - | ✓ | - | ✓ | YES |
| NAGphormer (*ICLR 2023*) | $O(n^3 + e)$ | $O(n)$ | ✓ | - | - | - | NO |
| Exphormer (*ICML 2023*) | $O(n^3)$ | $O(n + e)$ | ✓ | - | - | ✓ | Appr. |
| GOAT (*ICML 2023*) | $O(n \log(n))$ | $O(n + e)$ | ✓ | ✓ | - | - | Appr. |
| SGFormer (*NeurIPS 2023*) | - | $O(n + e)$ | - | - | - | ✓ | YES |
| Polynormer (*ICLR 2024*) | - | $O(n + e)$ | - | - | - | ✓ | YES |
| GoBFormer (*ICML 2024*) | $O(n \log(n))$ | $O(n^{\frac{4}{3}} + e)$ | - | ✓ | - | - | Appr. |
| DUALFormer (Ours) | - | $O(n + e)$ | - | - | - | - | YES |

## 3.3 COMPARISON WITH OTHER GTs.

This subsection presents a comparison between the proposed DUALFormer and some state-of-the-art GTs, aiming to demonstrate the advantages of DUALFormer. As shown in Table 1, DUALFormer has a more streamlined and efficient architectural design compared to the baselines. Taking this into account across three aspects, the detailed explanation is as follows.

**Scalability.** GraphTrans Wu et al. (2021) and SAT Chen et al. (2022) have a complexity of $O(n^2 + e)$ due to the computational demands of the self-attention mechanism, which calculates the full-pair attention scores between nodes, and the message-passing mechanism (in Eq. 1). This computational intensity becomes a bottleneck for scaling to large-scale graphs. In contrast, the proposed DUALFormer offloads such computationally intensive self-attention mechanism to the low-dimensional feature dimension, thus achieving linear time complexity. Besides, the absence of additional parameter requirements precludes an increase in spatial complexity. Both aspects ensure its scalability.

**Components.** To integrate the discriminative information of graph structure into their architectures, existing GTs often resort to the additional components: positional/structural encoding (*e.g.*, random-walk structural encoding and Laplacian eigenvectors encodings Rampásek et al. (2022)), augmented training loss (*e.g.*, edge regularization loss in NodeFormer), and the local GNNs, which necessitates additional parameters to control its effect (*e.g.*, a hyper-parameter in SGFormer and learnable parameters in Polynormer). Compared to them, DUALFormer features a more streamlined and efficient architectural design, requiring neither positional/structural encoding, augmented loss, nor additional parameters, and utilizes only simple local graph convolution and global attention.

**Expressivity.** Due to the limited receptive field of nodes, many existing GTs lack global expressivity. For example, NAGphormer achieves linear complexity through $k$-hop neighborhood sampling, yet this approach, while enhancing computation efficiency, leads to a compromise in the global expressivity. Additionally, several GTs (*e.g.*, Exphormer, GOAT, GoBFormer) attain linear complexity through clustering-like techniques; yet, this approach offers only an approximate global expressivity due to it equips nodes with an indirect global receptive field. In contrast, DUALFormer stands out by delivering both global expressivity and linear complexity. Section E.4 provides an analysis that demonstrates the superiority of the proposed self-attention module between features over the traditional self-attention module between nodes.

## 4 EXPERIMENTS

This section first evaluates the effectiveness and efficiency of the proposed DUALFormer by comparing its performance against various graph representation learning models on tasks of node classification and property prediction. Subsequently, it performs several additional experiments to provide a comprehensive understanding of DUALFormer. For details on the utilized benchmark datasets, the compared baselines, the experimental setups, and the tuned hyperparameters, refer to Section D.

Table 2: Accuracy in percentage (mean$_{\pm\text{std}}$) over 10 trials of the node classification task across seven graphs. Best and runner-up models are **bolded** and underlined, respectively.

| Model | Cora | CiteSeer | PubMed | Computers | Photo | CS | Physics |
|-------|------|----------|--------|-----------|-------|-----|---------|
| GCN | $81.60_{\pm0.40}$ | $71.60_{\pm0.40}$ | $78.80_{\pm0.60}$ | $89.65_{\pm0.52}$ | $92.70_{\pm0.20}$ | $92.92_{\pm0.12}$ | $96.18_{\pm0.07}$ |
| GAT | $83.00_{\pm0.70}$ | $72.10_{\pm1.10}$ | $79.00_{\pm0.40}$ | $90.78_{\pm0.13}$ | $93.87_{\pm0.10}$ | $93.61_{\pm0.14}$ | $96.17_{\pm0.08}$ |
| GraphSAGE | $82.68_{\pm0.47}$ | $71.93_{\pm0.85}$ | $79.41_{\pm0.53}$ | $91.20_{\pm0.29}$ | $94.59_{\pm0.14}$ | $93.91_{\pm0.13}$ | $96.49_{\pm0.06}$ |
| APPNP | $83.30_{\pm0.50}$ | $71.80_{\pm0.50}$ | $80.10_{\pm0.20}$ | $90.18_{\pm0.17}$ | $94.32_{\pm0.14}$ | $94.49_{\pm0.07}$ | $96.54_{\pm0.07}$ |
| SGC | $80.10_{\pm0.20}$ | $71.90_{\pm0.14}$ | $78.70_{\pm0.11}$ | $90.11_{\pm0.42}$ | $91.97_{\pm0.19}$ | $93.41_{\pm0.27}$ | $96.35_{\pm0.07}$ |
| GraphGPS | $82.84_{\pm1.03}$ | $72.73_{\pm1.23}$ | $79.94_{\pm0.26}$ | $91.19_{\pm0.54}$ | $95.06_{\pm0.13}$ | $93.93_{\pm0.12}$ | $97.12_{\pm0.19}$ |
| NodeFormer | $82.20_{\pm0.90}$ | $72.50_{\pm1.10}$ | $79.90_{\pm1.00}$ | $86.98_{\pm0.62}$ | $93.46_{\pm0.35}$ | $95.64_{\pm0.22}$ | $96.24_{\pm0.24}$ |
| NAGphormer | $82.12_{\pm1.18}$ | $71.47_{\pm1.30}$ | $79.73_{\pm0.28}$ | $91.22_{\pm0.14}$ | $95.49_{\pm0.11}$ | $\mathbf{95.75}_{\pm0.09}$ | $97.34_{\pm0.03}$ |
| Exphormer | $82.77_{\pm1.38}$ | $71.63_{\pm1.19}$ | $79.46_{\pm0.35}$ | $91.47_{\pm0.17}$ | $95.35_{\pm0.22}$ | $94.93_{\pm0.01}$ | $96.89_{\pm0.09}$ |
| GOAT | $83.18_{\pm1.27}$ | $71.99_{\pm1.26}$ | $79.13_{\pm0.38}$ | $90.96_{\pm0.90}$ | $92.96_{\pm1.48}$ | $94.21_{\pm0.38}$ | $96.45_{\pm0.28}$ |
| SGFormer | $84.50_{\pm0.80}$ | $72.60_{\pm0.20}$ | $80.30_{\pm0.60}$ | $91.99_{\pm0.76}$ | $95.10_{\pm0.47}$ | $94.78_{\pm0.20}$ | $96.60_{\pm0.18}$ |
| Polynormer | $83.25_{\pm0.93}$ | $72.31_{\pm0.78}$ | $79.24_{\pm0.43}$ | $\mathbf{93.68}_{\pm0.21}$ | $96.46_{\pm0.26}$ | $95.53_{\pm0.16}$ | $97.27_{\pm0.08}$ |
| CoBFormer | $84.71_{\pm0.73}$ | $74.29_{\pm0.51}$ | $81.42_{\pm0.53}$ | $92.21_{\pm0.41}$ | $95.46_{\pm0.49}$ | $94.91_{\pm0.07}$ | $97.33_{\pm0.25}$ |
| DUALFormer | $\mathbf{85.88}_{\pm0.10}$ | $\mathbf{74.45}_{\pm0.39}$ | $\mathbf{83.97}_{\pm0.43}$ | $93.16_{\pm0.17}$ | $\mathbf{96.74}_{\pm0.09}$ | $95.62_{\pm0.05}$ | $\mathbf{97.42}_{\pm0.03}$ |

**Datasets.** In the node classification experiments, seven benchmark datasets are employed, including *Cora* Sen et al. (2008), *CiteSeer* Sen et al. (2008), *PubMed* Sen et al. (2008), *Computers* Shchur et al. (2018), *Photo* Shchur et al. (2018), *CS* Shchur et al. (2018), and *Physics* Shchur et al. (2018). For the node property prediction, four benchmark datasets are utilized, including *ogbn-proteins* Hu et al. (2020), *ogbn-arxiv* Hu et al. (2020), *ogbn-products* Hu et al. (2020), and *pokec* Jure (2014). For details on the utilized datasets, refer to Section D.1.

**Baselines.** The baseline models include five typical GNNs: *GCN* Kipf & Welling (2016), *Graph-SAGE* Hamilton et al. (2017), *GAT* Velickovic et al. (2017), *APPNP* Klicpera et al. (2019), and *SGC* Wu et al. (2019), and eight GTs that represent the cutting-edge in this field: (*GraphGPS* Rampásek et al. (2022), *NodeFormer* Wu et al. (2022), *NAGphormer* Chen et al. (2023), *Exphormer* Shirzad et al. (2023), *GOAT* Kong et al. (2023), *SGFormer* Wu et al. (2024), *Polynormer* Deng et al. (2024), *CoBFormer* Xing et al. (2024)). Section D.2 offers an introduction to the compared models.

## 4.1 Experimental Results

**Node Classification.** The results of the models on the node classification task are presented in Tab. 2 and Fig. 2. Two observations can be drawn from Tab. 2. The first point is that the proposed DUAL-Former outperforms the baseline models on five of the seven datasets and delivers near-optimal performances on the remaining two. This demonstrates the superiority of DUALFormer. Moreover, the second point is that the proposed DUALFormer demonstrates significant performance advantages on Cora and PubMed datasets, exceeding all baselines by a wide margin. Specifically, the accuracies of DUALFormer exceed those of the runner-up model (CoBFormer) by $1.17\%$ and $2.55\%$, on these two datasets, respectively. Note that some of state-of-the-art baselines, *e.g.*, SGFormer, Polynormer, and CoBFormer, either explicitly or implicitly, incorporate GNNs, similar to DUALFormer. Thus, the performance advantages can be attributed to the fact that the proposed dual-dimensional architecture efficiently reduces information redundancy between the GNN and Self-Attention (SA) blocks, achieving a collaborative propagation of local and global information. Furthermore, as depicted in Fig. 2, the proposed DUALFormer not only achieves superior performance but also consumes less computation time compared to the GT baselines except SGFormer. To be specific, the red circles, representing the proposed DUALFormer, are superior left and above most other circles in the figure. This implies DUALFormer is lightweight, aligning with conclusions in Section 3.3. Besides, DUALFormer introduces modest memory usage, which indicates its potential for scalability.

**Node Property Prediction.** This experiment is designed to evaluate the scalability and effectiveness of the models by testing them on four large-scale graph datasets. The experiment results for the node property prediction task on these datasets are presented in Tab. 3. Upon observing this table, two key conclusions can be drawn. (1) Most GTs consistently show superior performance over the backbone GNNs across all datasets. This is because most GTs incorporate GNNs as their local module in architecture. Based on the concept of feature fusion, the integrated architecture typically achieves performance that is at least on par with its individual components. (2) Compared to the baselines, the proposed DUALFormer achieves superior performances across three of the four datasets. These

Table 3: Results (ROC-AUC for the ogbn-proteins and Accuracy for the others) in percentage (mean$_{\pm\text{std}}$) over 10 trials of node property prediction across four large-scale graphs. Best and runner-up models are **bolded** and underlined, respectively. #OOM means out of memory.

| Model | ogbn-proteins | ogbn-arxiv | ogbn-products | pokec |
|---|---|---|---|---|
| GCN | $72.51_{\pm0.35}$ | $71.74_{\pm0.29}$ | $75.64_{\pm0.21}$ | $75.45_{\pm0.17}$ |
| GAT | $72.02_{\pm0.44}$ | $71.95_{\pm0.36}$ | $79.45_{\pm0.59}$ | $72.23_{\pm0.18}$ |
| GraphSAGE | $77.68_{\pm0.20}$ | $71.49_{\pm0.27}$ | $78.50_{\pm0.14}$ | $75.63_{\pm0.38}$ |
| GraphGPS | $76.83_{\pm0.26}$ | $70.97_{\pm0.41}$ | $75.39_{\pm0.17}$ | $74.42_{\pm0.26}$ |
| NodeFormer | $77.45_{\pm1.15}$ | $67.19_{\pm0.83}$ | $72.93_{\pm0.13}$ | $71.00_{\pm1.30}$ |
| NAGphormer | $73.61_{\pm0.33}$ | $70.13_{\pm0.55}$ | $73.55_{\pm0.21}$ | $76.59_{\pm0.25}$ |
| Exphormer | $74.58_{\pm0.26}$ | $72.44_{\pm0.28}$ | $76.96_{\pm0.05}$ | $75.62_{\pm0.40}$ |
| SGFormer | $79.53_{\pm0.38}$ | $72.63_{\pm0.13}$ | $74.16_{\pm0.31}$ | $73.76_{\pm0.24}$ |
| Polynormer | $78.97_{\pm0.47}$ | $73.46_{\pm0.16}$ | $83.82_{\pm0.11}$ | **$86.10_{\pm0.05}$** |
| CoBFormer | $78.59_{\pm0.21}$ | $73.17_{\pm0.18}$ | $78.15_{\pm0.07}$ | $79.82_{\pm0.29}$ |
| DUALFormer | **$82.98_{\pm0.51}$** | **$73.71_{\pm0.22}$** | **$83.91_{\pm0.23}$** | $82.97_{\pm0.43}$ |

results demonstrate, on one hand, that DUALFormer is scalable to large-scale graphs and, on the other, illustrate its effectiveness. Notably, DUALFormer achieves a $3.45\%$ higher ROC-AUC over the second-ranking model SGFormer on the ogbn-proteins dataset, which highlights its superiority.

## 4.2 ADDITIONAL EXPERIMENTS

**Scalability Study.** This experiment aims to thoroughly examine the scalability of DUALFormer. It achieves this by randomly sampling nodes for the training set, gradually increasing the sample size from $10K$ to $100K$ nodes, and monitoring both training time and GPU memory usage as the dataset size varies. The variations in terms of running time and GPU memory consumption on the obg-products dataset are reported in Fig. 3(left) and (right), respectively. It is evident that both the running time and memory consumption of DUALFormer increase linearly with the size of the graph, indicating that DUALFormer possesses linear time and space complexity. This is consistent with the conclusion presented in Section 3.3, thereby demonstrating the scalability of DUALFormer.

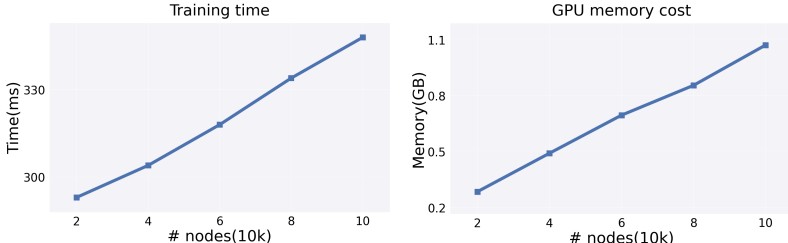

Figure 3: Training time and GPU memory usage of DUALFormer on the ogbn-products dataset.

**Backbone Performance Evaluation.** This experiment is designed to evaluate the impact of different backbone GNNs (including GCN, APPNP, and SGC) on model performances. Fig. 4 provides the experiment results, from which one can draw three conclusions. Firstly, all variants of DUAL-Former perform consistently across three datasets, suggesting that the performance of DUALFormer is not sensitive to different backbones. In particular, the most significant performance difference is observed on the CiteSeer dataset, with the original DUALFormer that equips SGC showing just a $1.29\%$ improvement over its variant that employs GCN. Secondly, DUALFormer and its variants achieve significantly improved performance over the baseline GNNs. For example, on the Cora dataset, the DUALFormer enhances the performance of GCN by $3.89\%$, and the variant with APPNP boosts the performance of APPNP by $2.83\%$. This underscores the effectiveness of DUALFormer's design. Finally, the performance of variants that pair with top-performing GNNs is generally high, indicating the potential for further improvements through integration with advanced GNNs.

**Parameter Sensitivity Analysis.** These experiments are performed to provide an intuitive understanding of hyperparameter selection. Firstly, as depicted in Fig. 5, which reports the performance variance for varying the dimension $d$, DUALFormer achieve consistently stable performances across

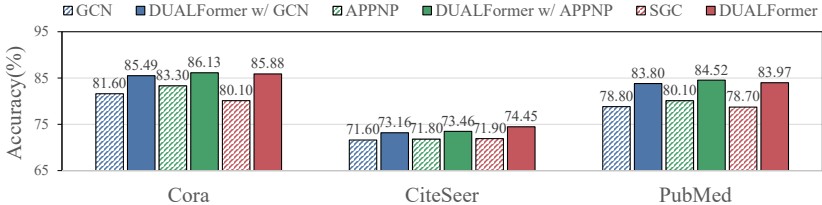

Figure 4: Comparison of DUALFormer with the employed backbone GNNs and the impact of different backbone GNNs on the model performances. The legend DUALFormer w/ GCN represents the variant of DUALFormer that incorporates GCN as its GNN block.

$\{32, 64, 128, 256\}$. In specific, the performance variation on these datasets remains minimal, staying within a $2.5\%$ margin. Therefore, DUALFormer has low sensitivity to the parameter $d$. In addition, DUALFormer does not always necessitate a high value for the parameter $d$ to function optimally; even a value as low as $64$ is sufficient. Secondly, it can be observed from Fig. 6 that DUALFormer exhibits a moderate sensitivity to the number of layers for both GNN and SA. To be specific, within the range of $\{1, 2, 3, 4, 5, 6, 7\}$, its performance tends to decline when the number of layers exceeds $3$. However, on the Citeseer dataset, where this effect is most pronounced, the performance decrease did not exceed $3.1\%$. This illustrates the stability of DUALFormer.

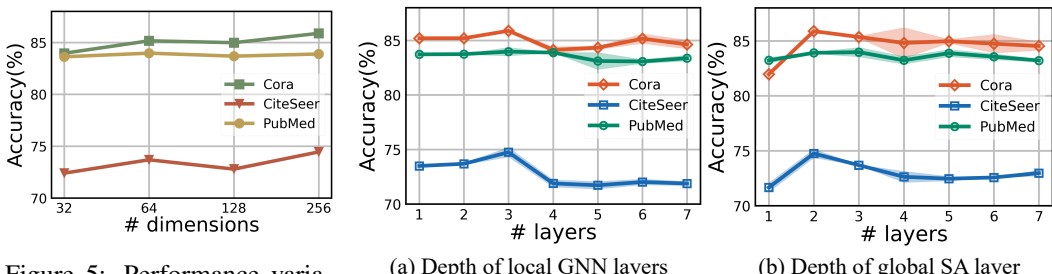

Figure 5: Performance variations for varying dimension $d$.

(a) Depth of local GNN layers

(b) Depth of global SA layer

Figure 6: Impact of local GNN and global SA layer depths

## 5 CONCLUSIONS

This paper presents DUALFormer, a straightforward and lightweight Graph Transformer (GT) architecture that utilizes a dual-dimensional design to enable efficient and effective learning of graph representations. DUALFormer addresses two key issues in existing GTs: the scalability issue on large-scale graphs and the tradeoff dilemma between local and global information. Extensive experiments across eleven graph benchmark datasets demonstrate the superiority of DUALFormer compared to existing Graph Neural Networks (GNNs) and GTs. Nonetheless, it still has scope for improvement. For example, 1) the universal GT architecture for homophilic and heterophilic graphs. Despite their global expressivity, GTs do not outperform well-designed GNNs on heterophilic graphs. Therefore, this is a research-worthy topic. 2) designing an edge-level GT architecture for edge-level tasks. Currently, GTs focus on graph-level and node-level tasks, with limited exploration in edge-level tasks. This is an open research question that merits further exploration.

## 6 ACKNOWLEDGMENTS

This work is supported in part by the National Natural Science Foundation of China (No. U22B2036, 62376088, 62276187, 62102413, 62272020), in part by the Hebei Natural Science Foundation (No. F2024202047, F2024202068), in part by the National Science Fund for Distinguished Young Scholarship (No. 62025602), in part by the Science Research Project of Hebei Education Department (BJK2024172), in part by the Guangxi Key Laboratory of Machine Vision and Intelligent Control (2023B03), in part by the Hebei Yanzhao Golden Platform Talent Gathering Programme Core Talent Project (Education Platform) (HJZD202509), in part by the Post-graduate's Innovation Fund Project of Hebei Province (CXZZBS2025036), in part by the XPLORER PRIZE, and in part by the Open Topics from the Lion Rock Labs of Cyberspace Security (under the project #LRL24009).

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

## A  ALGORITHM DESCRIPTION

The specifics of the DUALFormer architecture are presented in Algorithm 1, whereas the innovative global attention module is elaborated in Algorithm 2.

---

**Algorithm 1:** PyTorch-style Code for DUALFormer

```
# n: the number of nodes
# m: the number of edges
# f: the dimension of node attributes
# d: the dimension of hidden layers
# l_1: the number of local graph convolution layers
# l_2: the number of global attention layers

# x: input node attribute matrix with shape [n, f]
# edge_index: input graph structure with shape [2, m]
# alpha: combine rate of residual and output of each trans_layer
# local_convs: local graph convolution layers (e.g., SGConv from PyG)
# global_trans: global attention layers (implemented in Code 2)

residual_trans_list = [ ]

#  input projection layer
x = intput_project(x)
residual_trans_list.append(x)

#  global attention module
for trans_layer in enumerate(global_trans):
    x = trans_layer(x)
    x = alpha * x + (1-alpha) * residual_trans_list[-1]
    residual_trans_list.append(x)

#  local graph convolution module
residual_convs_list = [ ]
residual_convs_list.append(x)
for conv_layer in enumerate(local_convs):
    x = convs(x, edge_index)
    x = x + residual_conv_list[-1]
    residual_conv_list.append(x)

output = output_project(x)

#  negative log-likelihood loss calculation
y_pred = F.log_softmax(output, dim=1)
loss = criterion(y_pred[train_mask_idx], y_true[train_mask_idx])
```

---

## B  ADDITIONAL PRELIMINARIES

**Typical Graph Neural Networks (GNNs).** From the mentioned message-passing paradigm, typical GNNs such as GCN Kipf & Welling (2016), SGC Wu et al. (2019), APPNP Klicpera et al. (2019) employ the average function to implement the operations $\mathrm{AGG}(\cdot)$ and $\mathrm{COM}(,)$, that is,

$$GCN(\mathbf{A}, \mathbf{H}^l): \quad \mathbf{H}^{l+1} = \sigma(\tilde{\mathbf{A}}\mathbf{H}^l\mathbf{W}^l), \tag{10}$$

$$SGC(\mathbf{A}, \mathbf{H}^l): \quad \mathbf{H}^{l+1} = \tilde{\mathbf{A}}\mathbf{H}^l, \tag{11}$$

$$APPNP(\mathbf{A}, \mathbf{H}^l): \quad \mathbf{H}^{l+1} = (1 - \alpha) \times \tilde{\mathbf{A}}\mathbf{H}^l + \alpha \times \mathbf{H}^0, \tag{12}$$

where $\sigma$ denotes the non-linear activation function (e.g., ReLU($\cdot$)). $\tilde{\mathbf{A}} = (\mathbf{D}+\mathbf{I}_n)^{-\frac{1}{2}}(\mathbf{A}+\mathbf{I}_n)(\mathbf{D}+\mathbf{I}_n)^{-\frac{1}{2}}$ represents the normalized adjcency matrix, where $\mathbf{D}$ is the diagonal degree matrix with $d_{i,i} = 1 + \sum_j a_{i,j}$. These GNNs are employed in the proposed framework to capture the structure biases.

**Graph Transformers (GTs).** Given the node representations from the GNN, denoted as $\mathbf{H}$, and the SA module, denoted as $\mathbf{Z}$, the final representations $\mathbf{P}$ are produced according to two strategies: ① local-and-global fusion and ② local-to-global fusion. These can be formulated as

$$① \ \mathbf{P} \triangleq \text{Combination}(\mathbf{H}, \mathbf{Z}), \tag{13}$$

$$② \ \mathbf{P} \triangleq SA(\mathbf{H}), \ \mathbf{H} \triangleq GNN(\mathbf{A}, \mathbf{X}), \tag{14}$$

where $\text{Combination}(,)$ represents a combination function, such as $\mathbf{P} = \lambda\mathbf{H} + (1 - \lambda)\mathbf{Z}$ (*i.e.*, the linear weighting in SGFormer, and $\lambda$ is a parameter to balance these two terms).

## C  THEORETICAL PROOFS

To facilitate the proof, the node feature $\mathbf{x}_v \in \mathbb{R}^f$ is handled as a column vector. Assume that the node feature $\mathbf{x}_v$ and label $\mathbf{y}_v$ are the observations of random variables $\mathbb{X} = [\mathbb{X}_1, \ldots, \mathbb{X}_f]^\top$ and $\mathbb{Y}$ respectively. The node features are assumed to be mean-centered, namely, $\mathbb{E}[\mathbb{X}] = 0$. Note that the variance of random vector is the sum of the variance of each dimension, which can be represented as the trace of its covariance matrix. According to the law of total variance, the variance of $\mathbb{X}$ can be decomposed into intra-class and inter-class variances, that is,

$$\text{Var}[\mathbb{X}] = \underbrace{\mathbb{E}[\text{Var}[\mathbb{X}|\mathbb{Y}]]}_{\text{Intra-class Variance}} + \underbrace{\text{Var}[\mathbb{E}[\mathbb{X}|\mathbb{Y}]]}_{\text{Inter-class Variance}}, \tag{15}$$

where $\text{Var}[\mathbb{X}|\mathbb{Y} = k]$ stands for the variance of class $k$ and $\mathbb{E}[\mathbb{X}|\mathbb{Y} = k]$ denotes the center of class $k$. The conditional expectation of node feature $\mathbb{X}_j$ *w.r.t.* the $K$ classes can be represented as

$$\mathbf{e}_j = \{\mathbb{E}[\mathbb{X}_j|\mathbb{Y} = 1], \cdots, \mathbb{E}[\mathbb{X}_j|\mathbb{Y} = K]\} \in \mathbb{R}^K. \tag{16}$$

The key component of the Global Attention (GA) Module (in Eq. 8) is $\mathbf{XM}$, where the attention score matrix between features, *i.e.*, $\mathbf{M} = [m_{ij}] = \text{softmax}(\frac{(\mathbf{Q})^\top\mathbf{K}}{\sqrt{n}})$, is a learnable stochastic matrix. The node feature after GA is $\mathbf{M}^\top\mathbf{x}_v \in \mathbb{R}^f$, and thus the corresponding random variable is $\mathbf{M}^\top\mathbb{X}$. Then, the conditional expectation of node feature after GA is $\hat{\mathbf{e}}_j = \sum_i m_{ij}\mathbf{e}_i$. The quality of this attention score matrix between features $\mathbf{M}$ impacts this conditional expectation. Intuitively, a good attention score matrix should contain large elements, whose indices correspond to attributes with similar conditional expectations, *i.e.*, $\|\mathbf{e}_i - \mathbf{e}_j\|_2 \leq \varepsilon$.

Theorem 1 can be divided into the following two lemmas.

**Lemma 1.** *The intra-class variance of $\mathbf{M}^\top\mathbb{X}$ is less than or equal to that of $\mathbb{X}$, that is,*

$$\mathbb{E}[\text{Var}[\mathbf{M}^\top\mathbb{X}|\mathbb{Y}]] \leq \mathbb{E}[\text{Var}[\mathbb{X}|\mathbb{Y}]]. \tag{17}$$

**Lemma 2.** *If the learned attention score matrix is good enough that $\|\mathbf{e}_i - \mathbf{e}_j\|_2 \leq \varepsilon$ for any $m_{ij} \neq 0$, then the distance of conditional expectations before and after GA is also less than or equal to $\varepsilon$, i.e.,*

$$\|\hat{\mathbf{e}}_j - \mathbf{e}_j\|_2 \leq \varepsilon \tag{18}$$

*and $\varepsilon$ can be arbitrarily small with a proper $\mathbf{M}$.*

The proofs for these lemmas refer to Sections C.1 and C.2, respectively.

## C.1 PROOF FOR LEMMA 1

*Proof.* By denoting $\boldsymbol{\sigma} \in \mathbb{R}^m, \sigma_i \triangleq \sqrt{\mathrm{Var}[\mathbb{X}_i|\mathbb{Y} = k]}$ as the vector of intra-class variances, it can be proved that $\mathrm{Var}[\mathbf{M}^\top \mathbb{X}|\mathbb{Y} = k] \leq \mathrm{Var}[\mathbb{X}|\mathbb{Y} = k]$ as follows.

$$\mathrm{Var}[\mathbf{M}^\top \mathbb{X}|\mathbb{Y} = k] \tag{19}$$

$$= \mathrm{Tr}(\mathrm{Cov}(\mathbf{M}^\top \mathbb{X}|\mathbb{Y} = k)) \tag{20}$$

$$= \mathrm{Tr}(\mathbf{M}^\top \mathrm{Cov}(\mathbb{X}|\mathbb{Y} = k)\mathbf{M}) \tag{21}$$

$$= \mathrm{Tr}(\mathrm{Cov}(\mathbb{X}|\mathbb{Y} = k)\mathbf{M}\mathbf{M}^\top) \tag{22}$$

$$= \sum_{ij} \mathrm{Cov}(\mathbb{X}_i, \mathbb{X}_j|\mathbb{Y} = k)(\mathbf{M}\mathbf{M}^\top)_{ij} \tag{23}$$

$$\leq \sum_{ij} \sqrt{\mathrm{Var}[\mathbb{X}_i|\mathbb{Y} = k]}\sqrt{\mathrm{Var}[\mathbb{X}_j|\mathbb{Y} = k]}\,(\mathbf{M}\mathbf{M}^\top)_{ij} \tag{24}$$

$$= \sum_{ij} \sigma_i \sigma_j (\mathbf{M}\mathbf{M}^\top)_{ij} \tag{25}$$

$$= \boldsymbol{\sigma}^\top \mathbf{M}\mathbf{M}^\top \boldsymbol{\sigma} \tag{26}$$

$$\leq \|\boldsymbol{\sigma}\|_2^2 \tag{27}$$

$$= \sum_i \mathrm{Var}[\mathbb{X}_i|\mathbb{Y} = k] \tag{28}$$

$$= \mathrm{Var}[\mathbb{X}|\mathbb{Y} = k]. \tag{29}$$

Note that the second inequality holds since the eigenvalues of $\mathbf{M}$ are no more than 1. According to this, Lemma 1 can be obtained as follows.

$$\mathbb{E}[\mathrm{Var}[\mathbf{M}^\top \mathbb{X}|\mathbb{Y}]] \tag{30}$$

$$= \sum_k \mathrm{Pr}(\mathbb{Y} = k)\mathrm{Var}[\mathbf{M}^\top \mathbb{X}|\mathbb{Y} = k] \tag{31}$$

$$\leq \sum_k \mathrm{Pr}(\mathbb{Y} = k)\mathrm{Var}[\mathbb{X}|\mathbb{Y} = k] \tag{32}$$

$$= \mathbb{E}[\mathrm{Var}[\mathbb{X}|\mathbb{Y}]]. \tag{33}$$

$\square$

## C.2 PROOF FOR LEMMA 2

*Proof.*

$$\|\boldsymbol{e}_j - \widehat{\boldsymbol{e}}_j\|_2 = \left\|\boldsymbol{e}_j - \sum_i m_{ij}\boldsymbol{e}_i\right\|_2 \tag{34}$$

$$= \left\|\sum_i m_{ij}(\boldsymbol{e}_j - \boldsymbol{e}_i)\right\|_2 \tag{35}$$

$$\leq \sum_i m_{ij}\|\boldsymbol{e}_j - \boldsymbol{e}_i\|_2 \tag{36}$$

$$\leq \sum_i m_{ij}\varepsilon = \varepsilon. \tag{37}$$

The second inequality holds due to the Cauchy-Schwarz inequality. To show that there exists a matrix $\mathbf{M}$ such that $\varepsilon$ can be equal to 0, it need that $\mathbf{M}$ satisfying $\sum_i m_{ij}\boldsymbol{e}_i = \boldsymbol{e}_j$. Given that the number of features significantly exceeds the number of classes, the number of variables in the linear system is larger than that of equations. Thus, there exists an infinite number of solutions. Therefore, $\varepsilon$ can be arbitrarily small with a proper $\mathbf{M}$. $\square$

---

**Algorithm 2:** PyTorch-style Code for Global Attention Layer

```
# n: the number of nodes
# m: the number of edges
# d: the dimension of node features
# x: node features matrix with dimension [n, d]
# h: the number of head

# linear transformations
q = q_lins(x).reshape(-1, d, h)        #  [n, d, h]
k = k_lins(x).reshape(-1, d, h)        #  [n, d, h]
v = v_lins(x).reshape(-1, d, h)        #  [n, d, h]

# calculate attention score matrix with size [d, d]
qk = torch.einsum("lmh, ldh → mdh", q, k)
qk = torch.softmax(qk, dim=0)
trans_output = torch.einsum("lmh, mdh → ldh", v, qk)

# take average of all heads
trans_output = trans_output.mean(dim=-1)        #  [n, d]
```

---

## D EXPERIMENTAL DETAILS

### D.1 INTRODUCTION OF DATASETS

Table 4: Statistics for the eleven graph benchmark datasets used in Section 4, with seven employed for node classification tasks and the remaining four utilized for node property prediction tasks. For clarity, the "ogbn-" prefix is omitted for datasets including proteins, arxiv, and products.

|  | *Node Classification* | | | | | | | *Node Property Prediction* | | | |
|---|---|---|---|---|---|---|---|---|---|---|---|
|  | Cora | CiteSeer | PubMed | Computers | Photo | CS | Physics | proteins | arxiv | products | pokec |
| # Nodes | 2,708 | 3,327 | 19,717 | 13,752 | 7,650 | 18,333 | 34,493 | 132,534 | 169,343 | 2,449,029 | 1,632,803 |
| # Edges | 5,278 | 4,522 | 44,324 | 245,861 | 119,081 | 81,894 | 247,962 | 39,561,252 | 1,166,243 | 61,859,140 | 30,622,564 |
| # Features | 1,433 | 3,703 | 500 | 767 | 745 | 6,805 | 8,415 | 8 | 128 | 100 | 65 |
| # Classes | 7 | 6 | 3 | 10 | 8 | 15 | 5 | 2 | 40 | 47 | 2 |
| # Metric | ACC↑ | ACC↑ | ACC↑ | ACC↑ | ACC↑ | ACC↑ | ACC↑ | ROC-AUC↑ | ACC↑ | ACC↑ | ACC↑ |

Tab. 4 presents a compilation of statistical information for the eleven graph datasets (including Cora, CiteSeer, PubMed, Computers, Photo, CS, Physics, ogbn-proteins, ogbn-arxiv, ogbn-products, and pokec). Moreover, Tab. 7 offers statistical data for three graph datasets (including Roman-Empire, Question, and obgn-papers100M). These datasets are described as follows:

**Datasets for Node Classification.**

- Cora, CiteSeer, and PubMed Sen et al. (2008): They are three citation networks. The nodes signify scientific publications, characterized by attributes such as abstracts, keywords, full-text content, and derived features like a 0/1-valued word vector and TF/IDF weighted word vector. The edges represent citation relationships. The node labels correspond to research areas or topics.

- CS and Physics Shchur et al. (2018): They are two co-author networks. The nodes represent authors, with edges representing co-authorship relationships. The node attributes and labels indicate the keywords of the papers and the most active fields of study, respectively, for each author.

- Computers and Photo Shchur et al. (2018): They are two co-purchase networks from Amazon. The nodes represent products available for sale, which are characterized by attributes extracted from product reviews. The edges signify the co-purchase relationships. The node labels denote product categories or brands.

- Roman-Empire Platonov et al. (2023): It is a network constructed from the Roman Empire article on English Wikipedia. The nodes denote words in the article, with edges connecting sequential words or those with syntactic dependencies. The node attributes stand for FastText word embeddings Grave et al. (2018), and the node labels indicate syntactic roles.

- Question Platonov et al. (2023): This is a question-answering network. The nodes represent users, and the edges represent answer interactions between them. The node attributes are the average FastText embeddings Grave et al. (2018) of the user's description, plus a binary feature for users without descriptions. The node labels denote whether users are active.

**Datasets for Node Property Prediction.**

- ogbn-proteins, ogbn-arxiv, ogbn-products, and ogbn-papers100M: The are four large-scale graphs released recently by the Open Graph Benchmark (OGB) Hu et al. (2020). For the ogbn-product dataset, nodes represent products available for sale on Amazon; edges signify that the products are purchased together; node attributes are generated by extracting bag-of-words features from the product descriptions followed by the Principal Component Analysis (PCA) to reduce the dimension to 100. For the ogbn-proteins dataset, nodes denote individual proteins, and edges represent a variety of biologically significant interactions, including physical associations and co-expression patterns. For the ogbn-arxiv dataset, nodes represent papers on arXiv; edges stand for those papers cite other papers; the attributes of each node represent a 128-dimensional vector derived from averaging the embeddings of the words in the title and abstract of each paper. These word embeddings are generated using the skip-gram model over the MAG corpus Wang et al. (2020). For the ogbn-papers100M dataset, nodes correspond to academic papers from arXiv, characterized by 128-dimensional word embeddings as node attributes. Edges represent citation relationships between papers, and labels indicate the subject areas of the papers.

- pokec Jure (2014): This is an online social network in Slovakia. The nodes stand for users, with edges indicating friendships between users. The node attributes represent personal information, such as gender, age, hobbies, interests, and education. The node labels denote the attribute labels of the nodes, such as gender or age.

**Dataset Splitting.** To ensure the credibility and reproducibility of the experiment, the dataset splittings follow widely accepted schemes. To be specific, the split for the Cora, CiteSeer, and PubMed datasets follows the standard public strategies referenced in Kipf & Welling (2016), which allocate 20 nodes per class for training, 500 nodes for validation, and 1000 nodes for testing purposes. For the Computers, Photo, CS, and Physics datasets, the training, validation, and testing sets constitute $60\%$, $20\%$, and $20\%$ of the data, respectively. For the four datasets sourced from OGB (*i.e.*, ogbn-arxiv, ogbn-products, ogbn-proteins, and ogbn-papers100M), their standard public splits are utilized as referenced in Hu et al. (2020). For the pokec dataset, the partitioning follows Deng et al. (2024), with the training, validation, and testing sets distributed as $50\%$, $25\%$, and $25\%$, respectively. For the Roman-Empire and Questions datasets, the partitioning follows the scheme from Platonov et al. (2023), specifically, a $50\%/25\%/25\%$ split for training, validation, and testing.

### D.2    INTRODUCTION OF BASELINES

**Graph Neural Networks (GNNs).** The following outlines the GNN baselines used for comparison.

- GCN Kipf & Welling (2016): This is a seminal graph convolution network that integrates node attributes with topology structure through localized, diffusion-based convolutions.

- GAT Velickovic et al. (2017): This is a groundbreaking graph attention network that incorporates the popular attention mechanism into the graph convolution networks.

- GraphSAGE Hamilton et al. (2017): This is a scalable variant of GCN, which updates node features by sampling neighbor nodes and applying a learnable aggregation function.

- APPNP Klicpera et al. (2019): This is a variant of GCN that mitigates over-smoothing by aggregating node features based on personalized PageRank.

- SGC Wu et al. (2019): This is a simplified version of GCN that iteratively applies multiple feature aggregations to learn node embeddings.

- ChebNetII He et al. (2022): This is a spectral graph convolution network that enhances graph convolution through Chebyshev polynomial interpolation.
- OptBasisGNN Guo & Wei (2023): This is a spectral graph convolution network that computes the optimal polynomial basis for a given graph structure and graph signal.

**Graph Transformers (GTs).** The description of the GT baseline model is as follows.

- GraphGPS Rampásek et al. (2022): This is a versatile GT architecture, which encompasses position encoding, local message passing mechanism, and self-attention mechanism.
- NodeFormer Wu et al. (2022): This is a scalable GT architecture that utilizes a kernelized Gumbel-softmax operator for efficient and differentiable learning of graph structures.
- NAGphormer Chen et al. (2023): This is a scalable GT architecture, which utilizes multi-hop neighborhood aggregation to construct the token vectors for input sequences.
- Exphormer Shirzad et al. (2023): This is a sparse GT architecture that incorporates local attention, extended attention, and global attention through virtual nodes.
- GOAT Kong et al. (2023): This is a general GT architecture that utilizes the k-means-based dimensionality reduction to linearize computational complexity and tackles the challenges of both homophilic and heterophilic graphs.
- SGFormer Wu et al. (2024): This is a streamlined GT architecture that enables efficient information aggregation by only utilizing a single-layer global attention and GNN network.
- Polynormer Deng et al. (2024): This is a polynomial expressive GT model with linear complexity, which leverages a local-to-global attention scheme to learn node representations.
- CoBFormer Xing et al. (2024): This is a novel GT architecture that employs self-attention mechanisms within and across clusters to address the Over-globalizing problem.

For the seven GNN baselines, *i.e.*, GCN, GAT, GraphSAGE, APPNP, SGC, ChebNetII, and OptBasisGNN, we use the public library PyTorch Geometric (PyG) Fey & Lenssen (2019) and source code for their implementation. For the eight GT baselines, *i.e.*, GraphGPS, NodeFormer, NAGphormer, Exphormer, GOAT, SGFormer, Polynormer, and CoBFormer, we leverage their source codes. The sources are outlined as

- GNNs: https://github.com/pyg-team/pytorch_geometric/tree/master/torch_geometric/nn/conv
- ChebNetII: https://github.com/ivam-he/ChebNetI
- OptBasisGNN: https://github.com/yuziGuo/FarOptBasis
- GraphGPS: https://github.com/rampasek/GraphGPS
- NodeFormer: https://github.com/qitianwu/NodeFormer
- NAGphormer: https://github.com/JHL-HUST/NAGphormer
- Exphormer: https://github.com/hamed1375/Exphormer
- GOAT: https://github.com/devnkong/GOAT
- SGFormer: https://github.com/qitianwu/SGFormer
- Polynormer: https://github.com/cornell-zhang/polynormer
- CoBFormer: https://github.com/null-xyj/CoBFormer

### D.3 EXPERIMENTAL SETUP

**Hyper-parameters.** The model employs a semi-supervised learning framework, in which the model performances on the validation set are leveraged to tune the hyperparameter selection. By default, the hyper-parameters are meticulously selected via a grid search strategy. For the node classification task, DUALFormer is trained utilizing an Adam optimizer with the learning rate from $\{1e^{-3}, 1e^{-2}\}$ and the weight decay rate from $\{1e^{-5}, 1e^{-4}, 1e^{-3}\}$. The number of the local graph convolution layers and global attention layers are chosen from $\{1, 2, 3, 4, 5, 6, 7\}$, and the optimal results corresponding to these selections are depicted in Fig. 6. The dimensions of hidden layers are selected from $\{32, 64, 128, 256\}$ and the impacts are analyzed in Section 4.2. For each layer, the dropout

rate is among the set $\{0.1, 0.3, 0.5\}$. For the parameters unique to the attention layer: the number of heads is chosen from $\{2, 4\}$, and $\alpha$ of residual connection is selected from $\{0.1, 0.3, 0.5\}$. In addition, Batch Normalization and Layer Normalization are utilized as appropriate. For the node property prediction task on four large-scale graphs, a mini-batch training strategy is adopted. The value of $\alpha$ for the residual connection is chosen from the set $\{0.1, 0.2, 0.3, 0.4, 0.5\}$. The selection of hyper-parameters follows the compared baselines Chen et al. (2022); Wu et al. (2024); Deng et al. (2024). Refer to Tab. 5 for the parameter selections associated with the respective outcomes.

Table 5: Selected hyperparameters for DUALFormer.

| Dataset | GNN Layers | SA Layers | lr_GNN | lr_SA | Hidden Dim | $\alpha$ |
|---|---|---|---|---|---|---|
| Cora | 2 | 3 | 0.001 | 0.001 | 256 | 0.1 |
| CiteSeer | 2 | 3 | 0.001 | 0.01 | 256 | 0.1 |
| PubMed | 3 | 3 | 0.001 | 0.001 | 256 | 0.1 |
| Computers | 1 | 1 | 0.001 | 0.01 | 64 | 0.5 |
| Photo | 1 | 1 | 0.01 | 0.001 | 64 | 0.1 |
| CS | 1 | 1 | 0.001 | 0.001 | 256 | 0.1 |
| Physics | 1 | 1 | 0.001 | 0.001 | 256 | 0.3 |
| ogbn-proteins | 2 | 1 | 0.001 | 0.001 | 32 | 0.5 |
| ogbn-arxiv | 6 | 1 | 0.001 | 0.001 | 256 | 0.2 |
| ogbn-products | 3 | 1 | 0.001 | 0.001 | 256 | 0.5 |
| pokec | 6 | 1 | 0.0005 | 0.0005 | 512 | 0.1 |

**Configurations.** All experiments are conducted on two Linux machines as below.

Table 6: Servers and environment.

| | Server 1 | Server 2 |
|---|---|---|
| OS | Linux 5.15.0-82-generic | Linux 5.15.0-78-generic |
| CPU | Intel(R) Core(TM) i7-12700K CPU @ 3.6GHz | Intel(R) Xeon(R) Platinum 8360Y CPU @ 2.40GHz |
| GPU | GeForce RTX 4090 | NVIDIA A800 80GB PCIe |

# E    ADDITIONAL EXPERIMENT RESULTS

## E.1    FURTHER COMPARATIVE PERFORMANCE ANALYSIS

This subsection aims to evaluate the universality and scalability of the proposed DUALFormer. To this end, we compare DUALFormer with the latest state-of-the-art GNN baselines, including Cheb-NetII and OptBasisGNN, and GTs with linear complexity, including NodeFormer, NAGphormer, and SGFormer, on two heterophilic graphs (*i.e.*, Roman-Empire and Question) and three large-scale graphs (*i.e.*, ogbn-papers100M, ogbn-arxiv, and pokec). The baselines are configured as per their original descriptions, and DUALFormer settings are consistent with those mentioned in Section D.3. The experiment results are shown in Tab. 7.

Observing Tab. 7 results in the following two conclusions. Firstly, compared to the state-of-the-art GNNs, *i.e.*, ChebNetII, and OptBasisGNN, the baseline GTs perform worse in most datasets. This is primarily attributed to the inadequate integration of the GNN module with the SA module in these baselines. Secondly, in comparison to these baselines, the proposed DUALFormer achieves consistent performance advantages across all datasets, which demonstrates its universality and scalability. The performance boost stems from avoiding information redundancy between the two modules and effectively extracting discriminative information.

## E.2    FURTHER ASSESSMENT OF EFFICIENCY AND SCALABILITY.

The running time and GPU memory usage for the three large-scale graphs are detailed in Tab. 8. To demonstrate the impact of components in their architectures, particularly their self-attention mechanisms, on the scalability, the common hyper-parameters remain uniform across all models. It can be

Table 7: Accuracy in percentage (mean$_{\pm\text{std}}$) of node classification or property prediction on two heterophilic graphs and three large-scale graphs. Best and runner-up models are **bolded** and underlined, respectively. − means out of memory or failing to be finished within an acceptable time budget.

|  | Roman-Empire | Question | ogbn-papers100M | ogbn-arxiv | pokec |
|---|---|---|---|---|---|
| # Nodes | 22,662 | 48,921 | 111,059,956 | 169,343 | 1,632,803 |
| # Edges | 32,927 | 153,540 | 1,615,685,872 | 1,166,243 | 30,622,564 |
| # Attributes | 300 | 301 | 128 | 128 | 65 |
| # Classes | 18 | 2 | 172 | 40 | 2 |
| ChebNetII | $74.64_{\pm0.39}$ | $74.41_{\pm0.58}$ | $67.18_{\pm0.32}$ | $72.32_{\pm0.23}$ | $82.33_{\pm0.28}$ |
| OptBasisGNN | $76.91_{\pm0.37}$ | $73.82_{\pm0.83}$ | $67.22_{\pm0.15}$ | $72.27_{\pm0.15}$ | $82.83_{\pm0.04}$ |
| NodeFormer | $74.29_{\pm0.75}$ | $74.48_{\pm1.32}$ | - | $67.19_{\pm0.83}$ | $71.00_{\pm1.30}$ |
| NAGphormer | $74.45_{\pm0.48}$ | $75.13_{\pm0.70}$ | - | $70.13_{\pm0.55}$ | $76.59_{\pm0.25}$ |
| SGFormer | $73.91_{\pm0.79}$ | $77.06_{\pm1.20}$ | $66.01_{\pm0.37}$ | $72.63_{\pm0.13}$ | $73.76_{\pm0.24}$ |
| DUALFormer | $\mathbf{77.31_{\pm0.17}}$ | $\mathbf{78.62_{\pm0.56}}$ | $\mathbf{67.59_{\pm0.28}}$ | $\mathbf{73.71_{\pm0.22}}$ | $\mathbf{82.97_{\pm0.43}}$ |

Table 8: Training time and GPU memory usage on three large graphs. The best model is **bolded** and the runner-up is underlined, respectively.

|  | ogbn-arxiv | | ogbn-products | | pokec | |
|---|---|---|---|---|---|---|
| Method | Time/Epoch(s) | Mem.(GB) | Time/Epoch(s) | Mem.(GB) | Time/Epoch(s) | Mem.(GB) |
| GraphGPS | 0.114 | 8.43 | 8.13 | 2.56 | 2.58 | 2.43 |
| NodeFormer | 0.089 | 2.88 | 4.43 | 0.76 | 1.46 | 0.75 |
| NAGphormer | 0.349 | 6.02 | 13.98 | 4.13 | 3.59 | 1.20 |
| Exphormer | 0.123 | 6.76 | 7.64 | 2.27 | 2.17 | 2.11 |
| GOAT | 2.021 | 1.42 | 17.33 | 1.18 | 6.76 | 1.12 |
| SGFormer | **0.033** | **0.98** | 3.70 | **0.50** | 1.25 | **0.44** |
| Polynormer | 0.047 | 4.08 | 4.17 | 1.36 | 1.54 | 1.22 |
| CoBFormer | 0.103 | 6.46 | 5.62 | 9.28 | 2.44 | 3.70 |
| DUALFormer | 0.034 | 1.17 | **3.03** | 0.54 | **1.06** | 0.46 |

observed that DUALFormer exhibits the shortest runtime and has the top-2 smallest memory usage compared to the other GTs. The result highlights the efficiency and scalability of DUALFormer.

In addition to the aforementioned results, we conduct a specific comparison of the proposed DUALFormer with NAGphormer, which boasts the lowest training complexity of $O(n)$, as detailed in Tab. 1. Note that while the preprocessing steps of NAGphormer, which involve obtaining structural encodings, are computationally intensive, this cost can be amortized over multiple training epochs. Therefore, it justifies a comprehensive comparison of overall run times. Specifically, the experiment in Tab. 9 compares DUALFormer with NAGphormer in terms of performance, total running time, and GPU usage on three datasets: AMiner-CS Feng et al. (2020), Reddit Hamilton et al. (2017), and Amazon2M. To ensure a fair comparison, this experiment follows the setup of NAGphormer, with the exception that the data is divided into the commonly accepted ratio of $50\%/25\%/25\%$ for training, validation, and testing, respectively.

Table 9: Accuracy (%), running time (s), and GPU memory usage (MB) on three large graphs. The best model is highlighted in **bolded**.

|  | AMiner-CS | | | Reddit | | | Amazon2M | | |
|---|---|---|---|---|---|---|---|---|---|
|  | Acc(%) | Time(s) | Mem.(MB) | Acc(%) | Time(s) | Mem.(MB) | Acc(%) | Time(s) | Mem.(MB) |
| NAGphormer | $69.14_{\pm0.12}$ | 491.70 | 1672 | $\mathbf{95.96_{\pm0.02}}$ | 437.85 | 1710 | $90.73_{\pm0.24}$ | 2,629.46 | 1786 |
| DUALFormer | $\mathbf{69.30_{\pm0.27}}$ | 223.43 | 1347 | $95.71_{\pm0.07}$ | 328.71 | 1364 | $\mathbf{91.17_{\pm0.15}}$ | 11,772.03 | 1380 |

From this table, the following three results can be observed. Firstly, on these three large-scale graphs, the proposed DUALFormer achieves comparable performance to the baseline NAGphormer, which emphasizes the scalability and effectiveness of DUALFormer. Secondly, the proposed DUALFormer has short total running times on these datasets except for Amazon2M and less GPU usage on three datasets compared to NAGpormer. The advantage of DUALFormer primarily stems from its elimi-

nation of the need for preprocessing to acquire structural encoding and storage, unlike NAGphormer, which requires such steps. The reason for the failure of DUALFormer on the Amazon2M is that this graph has the highest number of edges compared to the other two graphs, leading to extended computation times for the GNN module. Thus, this results in a longer overall processing time compared to NAGphormer without the GNN module, aligning with the results of the complexity analysis.

### E.3 ADDITIONAL PARAMETER ANALYSIS

This experiment is conducted to analyze the parameter sensitivity of the proposed DUALFormer to hyperparameter $\alpha$. The search range of this parameter is detailed in Section D.3. The experimental results for seven small-scale and four large-scale graphs are shown in Tab. 10 and Tab. 11, respectively. Firstly, from Tab. 10, it can be observed that DUALFormer is not sensitive to this parameter. Specifically, within the search range, the variation of classification accuracy does not exceed $0.8\%$. Furthermore, the same conclusion is supported by the data in Tab. 11, as the performance variations do not exceed $1\%$. Thus, there is no need to concentrate excessively on the precise value.

Table 10: Performance variations for varying parameter $\alpha$ on six small-scale graphs.

|  | Cora | CiteSeer | PubMed | Computers | Photo | CS | Physics |
|---|---|---|---|---|---|---|---|
| 0.1 | $85.88_{\pm 0.10}$ | $74.45_{\pm 0.39}$ | $83.97_{\pm 0.43}$ | $93.09_{\pm 0.14}$ | $96.74_{\pm 0.09}$ | $95.62_{\pm 0.05}$ | $97.37_{\pm 0.02}$ |
| 0.3 | $85.20_{\pm 0.12}$ | $73.69_{\pm 0.03}$ | $83.91_{\pm 0.07}$ | $93.14_{\pm 0.15}$ | $96.43_{\pm 0.07}$ | $95.38_{\pm 0.04}$ | $97.42_{\pm 0.03}$ |
| 0.5 | $85.35_{\pm 0.08}$ | $74.06_{\pm 0.06}$ | $83.89_{\pm 0.52}$ | $93.16_{\pm 0.17}$ | $96.39_{\pm 0.09}$ | $95.52_{\pm 0.05}$ | $97.39_{\pm 0.02}$ |

Table 11: Performance variations for varying parameter $\alpha$ on four large-scale graphs.

|  | ogbn-proteins | ogbn-arxiv | ogbn-products | pokec |
|---|---|---|---|---|
| 0.1 | $82.25_{\pm 0.25}$ | $73.30_{\pm 0.06}$ | $83.19_{\pm 0.77}$ | $82.97_{\pm 0.43}$ |
| 0.2 | $82.26_{\pm 0.31}$ | $73.71_{\pm 0.22}$ | $82.99_{\pm 0.81}$ | $82.50_{\pm 0.46}$ |
| 0.3 | $82.20_{\pm 0.38}$ | $73.49_{\pm 0.13}$ | $83.79_{\pm 0.54}$ | $82.68_{\pm 0.34}$ |
| 0.4 | $82.32_{\pm 0.67}$ | $73.44_{\pm 0.17}$ | $83.07_{\pm 0.35}$ | $82.81_{\pm 0.45}$ |
| 0.5 | $82.98_{\pm 0.51}$ | $73.52_{\pm 0.22}$ | $83.91_{\pm 0.23}$ | $81.98_{\pm 0.79}$ |

### E.4 LOCAL AND GLOBAL EXPRESSIVITY

**Localizing Property.** The following theoretical analysis elucidates that in the proposed DUAL-Former, although the GNN module is set after the global attention module, it is capable of capturing the locality of the graph. Firstly, from the perspective of graph learning, many classical GNNs (e.g., GCN and SGC) can be induced by optimizing the objective function Yang et al. (2021); Zhu et al. (2021), namely

$$\arg \min_{\mathbf{H}} tr(\mathbf{H}^\top \tilde{\mathbf{L}} \mathbf{H}) = \frac{1}{2} \sum_{v,u} \tilde{a}_{v,u} \| \mathbf{h}_v - \mathbf{h}_u \|_2^2 \tag{38}$$

where $\tilde{\mathbf{L}}$ denotes the Laplacian matrix of the normalized adjacent matrix $\tilde{\mathbf{A}}$, and $\mathbf{H}$ stands for the node features such as $\mathbf{H} = \mathbf{XW}$ in GCN and $\mathbf{H} = \mathbf{X}$ in SGC. This indicates that GNNs essentially learn local information through feature updates that are constrained by the graph topology.

From the above perspective, the GNN module in the proposed DUALFormer is equivalent to solving the above objective function with $\mathbf{H} = \mathbf{Z}$, where $\mathbf{Z}$ denotes the node features obtained from the self-attention on the dimension regarding features. Thus, even as a post-processing technique, the GNN module can ensure localizing property by leveraging graph topology to constrain the feature updates.

**Globalizing Property.** The capability of capturing global information stems from the approximate equivalence between $\mathrm{softmax}(\mathbf{QK}^\top)\mathbf{V}$ and $\phi(\mathbf{Q})\phi(\mathbf{K})^\top\mathbf{V}$. Note that the global information can be captured by the attention between nodes, *i.e.*, $\mathrm{softmax}(\mathbf{QK}^\top)$, in the previous GTs. Thus, according to the combination law for matrix multiplication, it holds that $(\mathbf{QK}^\top)\mathbf{V} = \mathbf{Q}(\mathbf{K}^\top\mathbf{V})$ and $\mathrm{softmax}(\mathbf{QK}^\top)\mathbf{V} \approx \phi(\mathbf{Q})(\phi(\mathbf{K})^\top\mathbf{V})$. Therefore, this paper tends to approximate the expensive node attention $(\mathbf{QK}^\top)$ via efficient feature attention $(\mathbf{K}^\top\mathbf{V})$ since $\mathrm{softmax}(\mathbf{QK}^\top)\mathbf{V} \approx \phi(\mathbf{Q})(\phi(\mathbf{K})^\top\mathbf{V})$. Therefore, the proposed DUALFormer can capture global information.

## F   DISCUSSION ON ATTENTION BETWEEN FEATURES

**Feature Attention vs. Node Attention.** From the experiment results in Section 4, the proposed feature attention can lead to better efficiency and performance compared to node attention. This subsection seeks to offer insights into the efficiency and effectiveness of feature attention by comparing it with typical node attention.

Firstly, the high efficiency of the feature attention stems from the fact that the dimension of feature vectors is significantly smaller than the number of nodes. This has been justified through complexity analysis in Section 3.3. Secondly, the proposed feature attention can result in better model performance than the node attention. This is because it alleviates the conflict between limited training data and the need to model large-scale, complex relations between entries (nodes). On the one hand, the node attention in the previous GTs needs the model parameters to be accurately trained to model the relations of $n^2$ pairs. Unfortunately, the training data on graphs is often too limited to train them accurately. On the other hand, the feature attention in the proposed DUALFormer only requires to model the relations of $f^2$ pairs, which is much less than $n^2$ pairs. Hence, the training requirement of the model parameters is not to be as high as in the node attention, and the training data with the same scale is sufficient. Therefore, the performance can be improved by easing the conflict between limited training data and modeling relations.

**A specific case.** Let us consider a specific case where the feature is one-dimensional. In this scenario, neither the node attention nor the proposed feature attention can gather information. Firstly, in the case of a one-dimensional feature, $\mathbf{V}\sigma(\mathbf{Q}\mathbf{K}^\top)$ in Eq. 8 simplifies to $\mathbf{V} \cdot a$, where $a$ is a scalar and $\mathbf{V} \in \mathbb{R}^n$. Thus, the proposed feature attention is unable to gather information. Furthermore, for the one-dimensional feature, $\phi(\mathbf{Q})\phi(\mathbf{K})^\top$ in the node attention module reduces to a rank-1 matrix, whose rows only differ from each other by a scalar factor since $\phi(\mathbf{Q})$ is a column vector and $\phi(\mathbf{K})^\top$ is a row vector. Thus, the aggregation patterns/coefficients for different nodes, represented by the rows of $\phi(\mathbf{Q})\phi(\mathbf{K})^\top$, only differ by the scalar factor. As a result, $\phi(\mathbf{Q})\phi(\mathbf{K})^\top\mathbf{V}$ degrades to the same aggregation pattern/coefficient for different nodes. Since the essence of aggregation is the different aggregation patterns/coefficients for different nodes, the node attention loses this characteristic for a one-dimensional feature. Therefore, in this special case, neither the node attention gathers information, like the feature attention. This case demonstrates the importance of multiple features for the transformer. Thus, the proposed DUALFormer is further justified by exploring the correlation among multiple features with an additional transformer.

