# OpenReview forum: "DUALFormer: Dual Graph Transformer"
_ICLR.cc/2025/Conference — ICLR 2025 Poster_

### Official Review · Reviewer_GjDD · 2024-10-24

**Soundness:** 3
**Presentation:** 3
**Contribution:** 2
**Rating:** 6
**Confidence:** 5

**Summary:**

This paper develop a new architecture based on GNNs and modified Transformers. The authors conduct expensive experiments as well as theoretical analysis to show the effectiveness of the proposed method.

**Strengths:**

1.  This paper is easy to follow.
2.  The authors provide the theoretical analysis.
3.  The results on various datasets seem to be promising.

**Weaknesses:**

1.  The comparison of efficiency study seems to be not reasonable.
2.  The key contributions of the proposed method are not clear.
3.  The complexity analysis of the proposed method seems to be wrong.&#x20;

**Questions:**

I have the following questions:
1.  As the authors claim in Eq. 13, the proposed method only captures the feature-to-feature correlations. In my opinion, it is not the global information on the graph since it is unable to capture the relations between nodes. Why do authors claim the proposed method can capture the global information on the graph?
2.  According to the paper, the efficiency is the most important contribution of the proposed method. I think the authors express this point in a wrong way. Firstly, the authors claim that the computational complexity of the proposed method is $O(n)$ which is obviously wrong. Based on Eq. 14, the calculation involves the adjacency matrix. Hence, the computational complexity of this part is $O(E)$ and it is cannot be ignored since $|E|>|N|$ （even $|E|>>|N|$ on some graphs). Then, the authors only compare the time cost of each epoch to demonstrate the efficiency which is not reasonable. I think the total training time cost is the most important metric to demonstrate the efficiency of a method. So, the authors should report the overall training cost of each method for efficiency study, especially on large-scale graphs.  Maybe authors can refer to the settings in NAGphormer. For instance, can the proposed method achieve more efficient and more powerful performance than NAGphormer on Aminer, Reddit and Amazon2M?
3.  As shown in Section 4.2,  DUALFormer relies on the sampling strategy to perform on large-scale graphs, just like advanced linear graph Transformers. Hence, I think the GPU memory comparison is questionable since it is largely related to the batchsize. Do authors set the same batch for each method?
4.  The analysis of the $\alpha$ is missing. According to Table 5, the performance of DUALFormer could be sensitive to the value of $\alpha$. So, the parameter analysis of $\alpha$ should be added into the experiment section.

---

> ### Author Response · Authors · 2024-11-15
> **Response to Reviewer GjDD (Part 1)**
>
> >Q1. The key contributions of the proposed method are not clear.
>
> R1. The key contribution of the proposed DUALFormer is to introduce self-attention on the feature dimension. Although the design is simple, it possesses the following three excellent characteristics.
>
> 1) **A scalable self-attention**. Due to the quadratic computational complexity of their self-attention to the node dimension, Vanilla GTs often encounter scalability issues. In contrast, self-attention in DUALFormer operates efficiently, with complexity linearly related to the size of the graph. It is designed to capture inter-feature correlations to approximate global inter-node dependencies. As a result, it is potentially scalable to large-scale graphs.
>
> 2) **Improvement of discriminability**. Rigorous theoretical analysis demonstrates the rationality behind this design of self-attention on a novel dimension in improving the discriminability of node representations.
>
> 3) **Comprehensive expressivity**. Due to the global self-attention operating on the feature dimension, it seamlessly integrates with the local GNN module without compromising their expressivity. Therefore, DUALFormer achieves an automatic trade-off between local and global expressivity.
>
> Furthermore, the proposed DUALFormer has achieved **state-of-the-art** performances on many tasks, including node classification and node property prediction.
>
>
>  ---
> >Q2. As the authors claim in Eq. 13, the proposed method only captures the feature-to-feature correlations. In my opinion, it is not the global information on the graph since it is unable to capture the relations between nodes. Why do authors claim the proposed method can capture the global information on the graph?
>
> R2. The capability of capturing global information stems from the approximate equivalence between $\operatorname{softmax}(\mathbf{Q}\mathbf{K}^T)\mathbf{V}$ and $\mathbf{Q}(\mathbf{K}^T\mathbf{V})$. The global information is captured by the attention between nodes, i.e., $\operatorname{softmax}(\mathbf{Q}\mathbf{K}^T)$, in previous graph transformers. According to the combination law for matrix multiplication, it holds that $(\mathbf{Q}\mathbf{K}^T)\mathbf{V} = \mathbf{Q}(\mathbf{K}^T\mathbf{V})$ and $\operatorname{softmax}(\mathbf{Q}\mathbf{K}^T)\mathbf{V} \approx \mathbf{Q}(\mathbf{K}^T\mathbf{V})$. Thus, this paper tends to approximate the expensive node-node attention $(\mathbf{Q}\mathbf{K}^T)$ via efficient feature-feature attention $(\mathbf{K}^T\mathbf{V})$ since $\operatorname{softmax}(\mathbf{Q}\mathbf{K}^T)\mathbf{V} \approx \mathbf{Q}(\mathbf{K}^T\mathbf{V})$. Thus, the proposed DUALFormer can capture global information.
>
>
> ---
> >Q3. The authors claim that the computational complexity of the proposed method is $O(n)$, which is obviously wrong. Based on Eq. 14, the calculation involves the adjacency matrix. Hence, the computational complexity of this part is $O(E)$, and it cannot be ignored since $|E| > |N|$(even $|E| \gg |N|$ on some graphs).
>
> R3. Thanks for pointing out this error. In the previous version, only the complexities of self-attention in **ALL** GTs are considered. We will add the time complexity of GNNs to the time complexity of corresponding methods, including GraphTrans, SAT, GraphGPS, NodeFormer, NAGphormer, Exphormer, GOAT, SGFormer, Polynormer, GoBFormer, and the proposed DUALFormer. The adjusted time complexity is shown in the following table, where $e$ denotes the number of edges.
>
> |  |GraphTrans| SAT| GraphGPS | NodeFormer  | NAGphormer | Exphormer | GOAT | SGFormer | Polynormer | GoBFormer| DUALFormer
> |:--------:|:--------:|:--------:|:--------:| :---------:|:--------:|:--------:| :--------:|:--------:| :--------:| :---------:|:--------:|
> |Pre-processing | - | $O(n^3)$ | $O(n^3)$ | - | $O(n^3+e)$ | $O(n^3)$ | $O(nlogn)$ | - | - | $O(nlogn)$ | -|
> | Training | $O(n^2+e)$ | $O(n^2+e)$ | $O(n+e)$ | $O(n+e)$  | $O(n)$| $O(n+e)$ | $O(n+e)$| $O(n+e)$| $O(n+e)$| $O(n^{\frac{4}{3}}+e)$ |$O(n+e)$|
>
> The table reveals that the computational complexity of the proposed DUALFormer is linearly proportional to the number of nodes and edges, demonstrating its efficiency. Note that the complexity of DUALFormer aligns with those of the existing scalable graph transformers, such as NodeFormer, SGFormer, and Polynormer, while it does NOT require complicated preprocessing. This highlights the scalability of the proposed DUALFormer.  Therefore, the conclusion of the high scalability of the proposed DUALFormer is not changed.

---

> ### Author Response · Authors · 2024-11-15
> **Response to Reviewer GjDD (Part 2)**
>
> >Q4. The authors should report the overall training cost of each method for efficiency study, especially on large-scale graphs. Maybe authors can refer to the settings in NAGphormer. For instance, can the proposed method achieve more efficient and more powerful performance than NAGphormer on Aminer, Reddit, and Amazon2M?
>
> R4. According to your advice, we have compared the training cost in terms of total running time (s) and GPU memory (MB) of the proposed DUALFormer and NAGphormer. The batch size is uniformly set to 2000. The total number of training epochs is set to 100. All shared configurations are set to the same to ensure fairness. The result is shown in the table below.
>
> |  | AMiner-CS | Reddit  | Amazon2M |
> |:--------:|:--------:| :---------:|:--------:|
> |#Nodes| 593,486 | 232,965 | 2,449,029 |
> |#Edges| 6,217,004 | 11,606,919 | 61,859,140 |
> |#Attributes| 100 | 602 | 100 |
> |#Classes| 18 | 41 | 47 |
> | |Accuracy(%) / Time(s) / Memory(MB) | Accuracy(%) / Time(s) / Memory(MB) | Accuracy(%) / Time(s) / Memory(MB) |
> | NAGphormer | 56.21$_{± 0.42}$ / 38.51 / 84 | 93.58$_{± 0.05}$ / 30.88 / 140   | 83.97$_{± 0.43}$ / 568.91 / 146 |
> | DUALFormer |  58.56$_{± 0.50}$ / 2.32 / 30 | 94.71$_{± 0.07}$ / 6.82 / 64  |84.80$_{± 0.22}$ / 40.38 / 26 |
>
> From the table, two results can be observed: firstly, the proposed DUALFormer consistently outperforms NAGpormer across the three datasets, and secondly, the proposed DUALFormer has short running times across the three datasets compared to the baseline NAGpormer. The advantage of DUALFormer primarily stems from its elimination of the need for preprocessing to acquire structural encoding and storage, unlike NAGphormer, which requires such steps. This agrees with the conclusion of the complexity analysis.
>
> ---
> >Q5. As shown in Section 4.2, DUALFormer relies on the sampling strategy to perform on large-scale graphs, just like advanced linear graph Transformers. Hence, I think the GPU memory comparison is questionable since it is largely related to the batch size. Do authors set the same batch for each method?
>
> R5. I understand your concern about fairness. The common hyper-parameters (including batch size) are the **same for each model**, as mentioned in Line 953. Specifically, all models are trained on ogbn-arxiv using full batch, whereas, for ogb-products and pokec, the batch size is set to 10K. We will explicitly mention this in the captions of the corresponding tables and figures in the revised manuscript.
>
> ---
> >Q6. The analysis of the $\alpha$ is missing. According to Table 5, the performance of DUALFormer could be sensitive to the value of $\alpha$. So, the parameter analysis of $\alpha$ should be added into the experiment section.
>
> R6. Thanks for your careful check. The impact of the hyper-parameter $\alpha$ on model performance is shown below.
>
> |  | Cora | CiteSeer  | PubMed | Computers | Photo  | CS | Physics |
> |:--------:|:--------:| :---------:|:--------:|:--------:| :---------:|:--------:|:--------:|
> | 0.1 | 85.88$_{± 0.10}$| 74.45$_{±0.39}$  |83.97 $_{± 0.43}$| 93.09$_{±0.14}$ |96.74$_{±0.09}$ |  95.62$_{±0.05}$| 97.37$_{± 0.02}$|
> | 0.3 |  85.20$_{± 0.12}$| 73.69$_{± 0.03}$ |83.91$_{± 0.07}$ | 93.14$_{±0.15}$ | 96.43$_{± 0.07}$| 95.38$_{± 0.04}$|  97.42$_{±0.03}$|
> | 0.5 | 85.35$_{± 0.08}$| 74.06$_{±0.06}$ | 83.89$_{± 0.52}$| 93.16$_{±0.17}$ |96.39$_{±0.09}$ | 95.52$_{± 0.05}$ |  97.39$_{± 0.02}$|
> | Margin | 0.68| 0.39| 0.08 | 0.07 | 0.35 | 0.24 | 0.05 |
>
> From the table, it can be observed that DUALFormer is not sensitive to the parameter $a$. Specifically, within the parameter selection range, the variation of classification accuracy does not exceed $0.7\%$.

---

> > ### Comment · Reviewer_GjDD · 2024-11-15
> >
> > I appreciate the authors for their detailed rebuttal. However, based on the current version, I do not believe the work is ready for publication. My concerns are as follows:
> >
> > First, as the authors claimed, the proposed method actually only capture the relations between each feature. As the results, it is efficient since the dimension of the feature vector is much smaller than the number of nodes. Maybe the results reported in this paper show that the feature-feature attention can lead to better model performance than the node-node attention.
> >
> > Secondly, the experimental results presented by the authors remain unclear and potentially misleading. For instance, in their response to Question 4, they mention a maximum GPU cost of 146 MB. I strongly recommend that the authors carefully review and validate their experimental setup and results to ensure they are accurate and reproducible.
> >
> > Lastly, the investigation of the hyperparameter $\alpha$ is insufficient. In the original version of the manuscript, Table 5 indicates that the optimal value of $\alpha$ for the proteins dataset is zero. This finding should be critically examined and explained. Furthermore, the authors state that the search space for $\alpha$ includes 0.1, 0.3, and 0.5, yet they report 0.2 as the optimal value for the arXiv dataset. These inconsistencies raise questions about the reliability and thoroughness of the experimental results.
> >
> > In light of these issues, I believe the current version of this work requires substantial revisions before it can be considered for acceptance. The current version lacks the necessary rigor and clarity to support the claims made, and a more meticulous examination of the experimental designs and results is warranted.

---

> > > ### Author Response · Authors · 2024-11-16
> > > **Respense to Reviewer GjDD**
> > >
> > > > Q1. As the authors claimed, the proposed method actually only capture the relations between each feature. As a result, it is efficient since the dimension of the feature vector is much smaller than the number of nodes. Maybe the results reported in this paper show that the feature-feature attention can lead to better model performance than the node-node attention.
> > >
> > > R1. You are right. Firstly, the high efficiency stems from the fact that the dimension of the feature vectors is significantly smaller than the number of nodes. This has been justified through complexity analysis.
> > > Secondly, feature-feature attention can lead to better model performance than node-node attention. This is because we ease the conflict between limited training data and modeling large-scale complicated relations between entries. On the one hand, node-node attention in the previous GT needs the model parameters to be accurately trained to model relations of $n^2$ pairs. However, the training data on graphs is often too limited to train them accurately. On the other hand, feature-feature attention in the proposed DUALFormer only requires to model the relations of $f^2$ pairs, which is much less than $n^2$ pairs. Thus, the training requirement of the model parameters is NOT to be as high as in node-node attention, and the training data with the same scale is sufficient. As a result, performance can be improved by easing the conflict between limited training data and modeling relations.
> > > Finally,  there are some key points we want to clarify.
> > > Firstly, the proposed DUALFormer, as its name indicates, consists of two complementary components, i.e.,  a GNN block for local information and a feature-feature self-attention (SA) block for global information, instead of only capturing relations between features.
> > > Secondly, the superiority of the proposed feature-feature is justified by both experiments and theoretical analysis. Theorem 1 demonstrates it can improve the discriminability, which ensures performance enhancement.
> > >
> > > ---
> > > > Q2. The experimental results presented by the authors remain unclear and potentially misleading. For instance, in their response to Question 4, they mention a maximum GPU cost of 146 MB. I strongly recommend that the authors carefully review and validate their experimental setup and results to ensure they are accurate and reproducible.
> > >
> > > R2. Thank you for your feedback. We acknowledge the error in our experimental setup and are correcting it to align with the protocol from "NAGFormer." We are rerunning the experiments with the widely accepted 50%/25%/25% data split. We are committed to accuracy in our research, and as such, we will provide a detailed report of the updated results and experimental details upon completion. We believe that these revisions will not only address your concerns but also strengthen the integrity of our paper.
> > >
> > > ---
> > > > Q3. The investigation of the hyperparameter \alpha is insufficient. In the original version of the manuscript, Table 5 indicates that the optimal value of \alpha for the proteins dataset is zero. This finding should be critically examined and explained. The authors state that the search space for \alpha includes 0.1, 0.3, and 0.5, yet they report 0.2 as the optimal value for the arXiv dataset.
> > >
> > > R3. Thanks for your meticulous review and for bringing the typographical error to our attention. Upon re-examining the section, we have identified the oversight and can confirm that the intended parameter value is 0.5 not the incorrectly stated 0.
> > >
> > > We acknowledge the confusion that arose from our failure to specify the parameter range for the node property prediction task in our initial submission. We would like to clarify that the mentioned parameter range {0.1, 0.3, 0.5} is intended for the node classification task, as stated in Line 912: "For the node classification task, ...". We have identified the correct experimental parameter range for the node property prediction task as {0.1, 0.2, 0.3, 0.4, 0.5}. We understand the importance of this detail and its impact on the interpretation of our results. In the revised manuscript, we will include the results obtained within this range and provide a sensitivity analysis.
> > >
> > > ---
> > > We are carefully incorporating the discussed points into the revised manuscript. This revised version, along with our rebuttal, will be submitted shortly. We sincerely appreciate your professional and valuable suggestions, as they have significantly contributed to enhancing the quality of our paper.

---

> > > > ### Comment · Reviewer_GjDD · 2024-11-16
> > > >
> > > > I thank the authors for their rebuttal. I have slightly raised my scores due the authors' sincerity. And I will further raise my score if the authors do carefully revise their paper based on the above discussions.

---

> > > > > ### Author Response · Authors · 2024-11-20
> > > > > **Response to Reviewer GjDD**
> > > > >
> > > > > We sincerely appreciate your professional and valuable feedback, which has significantly enhanced the quality of our paper. We would like to address each of your concerns individually in response.
> > > > >
> > > > > 1. **The performance, total running time, and GPU usage comparison between DUALFormer and NAGphormer on large graphs**. The results can be found in Tab. 9 of the revised manuscript. The results highlight the scalability and effectiveness of DUALFormer while revealing the drawbacks introduced by the GNN module.
> > > > >
> > > > > 2. **The analysis of the hyperparameter $\alpha$**. The search range is presented in Section C. 3. The sensitivity analysis of this hyperparameter is detailed in Section D. 3. The findings indicate that DUALFormer exhibits stability to variations in $\alpha$.
> > > > >
> > > > > We hope these rebuttals have alleviated your concerns regarding 1) the key contributions of our proposed method, 2) the global expressivity of DUALFormer, and 3) the efficiency and scalability of the model, 4) the effectiveness of the feature-feature attention. We are grateful for your expertise and have benefited greatly from our interactions. If you have any more questions or need clarification, please let us know. We look forward to further discussions with you.

---

> > > > > > ### Comment · Reviewer_GjDD · 2024-11-23
> > > > > >
> > > > > > Thanks for your response. I have carefully read the response and the revised manuscript. I think the authors have addressed my concerns. Hence, I raise my score to 6.

---

### Official Review · Reviewer_Q319 · 2024-10-27

**Soundness:** 3
**Presentation:** 4
**Contribution:** 3
**Rating:** 8
**Confidence:** 5

**Summary:**

This paper introduces DUALFormer, a graph transformer that tackles the challenges of the scalability and trade-off between local and global expressivity faced by current models. The motivation is to model the global dependencies among nodes by approximately characterizing the correlations between features. DUALFormer adopts a simple, intuitive design that includes local graph convolutional networks operating on the node dimension and a global self-attention mechanism operating on the feature dimension. The effectiveness and efficiency of the proposed DUALFormer are demonstrated in experimental evaluations across node classification and node property prediction tasks.

**Strengths:**

1) The motivation for the dual design of local and global modules in this paper is clear and interesting.
2) The model DUALFormer is simple and efficient with a solid theoretical foundation.
3) The paper offers extensive experimental validation across various datasets.
4) The paper is well-organized and easy to read.

**Weaknesses:**

1) The paper has some minor errors that need fixing. For example, Table 2 misses the mean value for the GraphGPS model on the Citeseer dataset.
2) To enhance readability, Equation 13 should be split into two or three equations.
3) The model DUALFormer places the GNN layers, such as the SGC layers, after the attention layers. What is the rationale behind this design? Is it possible to reverse this order?
4) Figure 4 shows that the model utilizing APPNP outperforms the one using SGC in the Cora and Pubmed datasets. What accounts for this performance difference?
5) The effect of certain hyper-parameters, such as the parameter $\alpha$ in Equation 13, on performance has yet to be unverified.
6) The paper does not mention any plans to open-source the code.

* Update after carefully reviewing the authors' responses: The authors have provided detailed and thoughtful replies that effectively address most of my concerns. At this stage, I am pleased to increase my evaluation of the paper to '8: accept, good paper'.

**Questions:**

Update after carefully reviewing the authors' responses: no further concerns

---

> ### Author Response · Authors · 2024-11-15
> **Response to Reviewer Q319**
>
> > Q1. The paper has some minor errors that need fixing. For example, Table 2 misses the mean value for the GraphGPS model on the Citeseer dataset.
>
> R1. Thanks for your careful checking. We will thoroughly check the manuscript to correct any omissions.
>
> ---
> > Q2. To enhance readability, Equation 13 should be split into two or three equations.
>
> R2. Based on your suggestion, we will divide Equation 13 into three formulas by row.
>
> ---
> > Q3. The model DUALFormer places the GNN layers, such as the SGC layers, after the attention layers. What is the rationale behind this design? Is it possible to reverse this order?
>
> R3. We would like to explain this design choice as follows.
>
> This choice is primarily motivated by the desire to decouple local and global modules, thereby minimizing their mutual interference. The self-attention module generally relies on input representations to calculate attention coefficients, whereas the GNN module, typically GCN and GAT, utilizes fixed propagation coefficients that are input-independent. Therefore, placing the GNN module after the self-attention module can mitigate their mutual interference and ensure that comprehensive information is retained. Thus, it seems that this order cannot be reversed.
>
> ---
> > Q4. Figure 4 shows that the model utilizing APPNP outperforms the one using SGC in the Cora and Pubmed datasets. What accounts for this performance difference?
>
> R4. This performance difference is primarily attributed to the difference in the localizing property of these two models. As can be seen in Figure 4, the original APPNP has a performance advantage over the original SGC on the Cora and PubMed datasets. This demonstrates the superiority of the former in terms of localizing property. By designing the global self-attention module in the pairwise dimension of the local GNN module, DUALFormer naturally obtains the global information with the guarantee that it does not interfere with each other. Thus, the DUALFormer based on APPNP with superior localizing property outperforms the DUALFormer based on SGC.
>
> Thank you for the reminder. It underscores the compatibility of DUALFormer and suggests the potential for further enhancements by integrating it with more advanced GNNs.
>
> ---
> > Q5. The effect of certain hyper-parameters, such as the parameter $\alpha$ in Equation 13, on performance has yet to be unverified.
>
> R5. Thanks for your careful check. The impact of the hyper-parameter $\alpha$ on model performance is shown below.
>
> |  | Cora | CiteSeer  | PubMed | Computers | Photo  | CS | Physics |
> |:--------:|:--------:| :---------:|:--------:|:--------:| :---------:|:--------:|:--------:|
> | 0.1 | 85.88$_{± 0.10}$| 74.45$_{±0.39}$  |83.97 $_{± 0.43}$| 93.09$_{± 0.14}$ |96.74$_{±0.09}$ |  95.62$_{±0.05}$| 97.37$_{± 0.02}$|
> | 0.3 |  85.20$_{± 0.12}$| 73.69$_{± 0.03}$ |83.91$_{± 0.07}$ | 93.14$_{± 0.15}$ | 96.43$_{± 0.07}$| 95.38$_{± 0.04}$|  97.42$_{±0.03}$|
> | 0.5 | 85.35$_{± 0.08}$| 74.06$_{±0.06}$ | 83.89$_{± 0.52}$| 93.16$_{± 0.17}$ |96.39$_{±0.09}$ | 95.52$_{± 0.05}$ |  97.39$_{± 0.02}$|
> | Margin | 0.68| 0.39| 0.08 | 0.07 | 0.35 | 0.24 | 0.05 |
>
> From the table, it can be observed that DUALFormer is not sensitive to the parameter $a$. Specifically, within the parameter selection range, the variation of classification accuracy does not exceed $0.7\%$.
>
> ---
> > Q6. The paper does not mention any plans to open-source the code.
>
> R6. We promise to open-source the code and provide a GitHub link once the paper is accepted.

---

> > ### Comment · Reviewer_Q319 · 2024-11-23
> > **I am inclined to increase my evaluation of the paper more favorably.**
> >
> > Thank you for the detailed response, which effectively addresses most of my concerns. The clarifications provided on the model design and hyperparameter selection were particularly helpful and have improved my understanding of the work. As a result, I am willing to increase my score for the paper.

---

### Official Review · Reviewer_CPJ4 · 2024-11-01

**Soundness:** 2
**Presentation:** 3
**Contribution:** 2
**Rating:** 6
**Confidence:** 3

**Summary:**

To address the scalability limitations of graph transformers (GTs) and the challenge of balancing local and global information, this paper introduces DualFormer, a novel GT architecture. DualFormer calculates global attention along the feature dimension, enabling the model to perform effectively and efficiently on large graphs while maintaining strong performance.

**Strengths:**

- The writing is generally clear and accessible, making the paper readable and easy to follow.
- The proposed method is both understandable and implementable, yet effective. It performs well on several datasets.
- The paper includes diverse experimental analyses, such as node classification, node property prediction, ablation studies, and parameter sensitivity analyses. Furthermore, the authors offer theoretical guarantees to support the method.

**Weaknesses:**

- The motivation for the study is not fully convincing. Further details are provided in the questions below.
- Since the paper emphasizes the method’s scalability, additional experiments on larger graphs would reinforce this claim. Suggested datasets include *Roman-Empire*, *Question[1]*, *Wiki*, and *ogbn-papers100M*. Moreover, the GNN baselines in Tables 2 and 3 are outdated, which may reduce the persuasiveness of the results. For instance, the statement, “Most GTs consistently show superior performance over GNNs across all datasets” (line 451), would be more convincing if compared with recent GNN baselines, such as *ChebNetII[2]* and *OptBasis[3]*, to present a more comprehensive evaluation.
- Minor Issues: There are a few typographical errors, such as "abov" (line 182). Consistent notation throughout the paper is also preferable. For instance, in line 168, there is a "$\times$" symbol between a scalar and a matrix, but not in line 216. Additionally, line 191 includes a "$\cdot$" between matrices, whereas line 167 does not.

[1] A critical look at the evaluation of GNNs under heterophily: Are we really making progress? In ICLR 2023.

[2] Convolutional Neural Networks on Graphs with Chebyshev Approximation, Revisited. In NeurIPS 2022.

[3] Graph Neural Networks with Learnable and Optimal Polynomial Bases. In ICML 2023.

**Questions:**

- The first question concerns the reasonableness of applying softmax to the global correlations between features.

  - In standard self-attention, $ \mathbf{O} = \exp(\text{sim}(\mathbf{Q}, \mathbf{K}))\mathbf{V} $ (Eq. 6).
  - Through linearized attention, $ \mathbf{O} = \phi(\mathbf{Q}) \phi(\mathbf{K})^\top \mathbf{V} $ (Eq. 11), where each element in $ \phi(\mathbf{Q}) \phi(\mathbf{K})^\top $ is non-negative, representing attention weights (global dependencies between nodes).
  - By the commutative property of matrix multiplication, $ \mathbf{O} = \phi(\mathbf{Q}) (\phi(\mathbf{K})^\top \mathbf{V}) $, so we can interpret $ (\phi(\mathbf{K})^\top \mathbf{V}) $ as a correlation matrix (with elements that can be positive or negative).

  However, in Eq. 13, $ \mathbf{V} \text{softmax}(\mathbf{Q}^\top \mathbf{K}) $, i.e., $ \mathbf{Q} \text{softmax}(\mathbf{K}^\top \mathbf{V}) $, differs from $ \phi(\mathbf{Q}) (\phi(\mathbf{K})^\top \mathbf{V}) $ because elements in $\text{softmax}(\mathbf{K}^\top \mathbf{V}) $ are all non-negative, unlike those in $ (\phi(\mathbf{K})^\top \mathbf{V})$. Could you clarify these differences and explain why it is reasonable to replace $ \phi(\mathbf{Q}) (\phi(\mathbf{K})^\top \mathbf{V}) $ with $ \mathbf{Q} \text{softmax}(\mathbf{K}^\top \mathbf{V}) $?

- The second question pertains to the interpretation of the proposed global attention. The method appears to aggregate information along the feature dimension, unlike previous approaches that gather global information across all or most nodes in a graph. For a one-dimensional feature, $ \mathbf{V}  \text{softmax}(\mathbf{Q} \mathbf{K}^T) $ in Eq. 13 reduces to $ \mathbf{V} \cdot \alpha $, where $ \alpha $ is a scalar and $ \mathbf{V} \in \mathbb{R}^{n} $. How can this be understood as gathering information from a global perspective?

---

> ### Author Response · Authors · 2024-11-15
> **Response to Reviewer CPJ4 (Part 1)**
>
> > Q1. Suggested datasets include Roman-Empire, Question[1], Wiki, and ogbn-papers100M.
>
> R1. According to your suggestion, we have conducted model comparisons on the Roman-Empire, Question, and ogbn-papers100M datasets. For the Roman-Empire and Questions datasets, the data partitioning follows the scheme from [1], specifically, a 50/25/25 split for training, validation, and testing. For the ogbn-papers100M, the split ratio is public split [2], namely 78/8/14. The statistics of the dataset and the experimental results are shown in the following table.
>
> |  | Roman-Empire | Question  | ogbn-papers100M |
> |:--------:|:--------:| :---------:|:--------:|
> |#Nodes| 22,662 | 48,921 | 111,059,956 |
> |#Edges| 32,927 | 153,540 | 1,615,685,872 |
> |#Attributes| 300 | 301 | 128 |
> |#Classes| 18 | 2 | 172 |
> | NAGphormer | 74.45$_{±0.48}$ |   75.13$_{±0.70}$ | - |
> | GOAT | 72.30$_{±0.48}$ | 75.95$_{±1.38}$  | - |
> | SGFormer |  73.91$_{±0.79}$ | 77.06$_{±1.20}$ | 66.01$_{±0.37}$ |
> | DUALFormer |  77.31$_{±0.17}$ | 78.62$_{±0.56}$ | 67.59$_{±0.28}$ |
>
> The table reveals that, in comparison to the baselines, our proposed DUALFormer achieves consistently performance advantages on all three datasets, underscoring its superiority and scalability.
>
> [1] A critical look at the evaluation of GNNs under heterophily: Are we really making progress? ICLR 2023
> [2] Open Graph Benchmark: Datasets for Machine Learning on Graphs. NeurIPS 2020
>
> ---
> >Q2. The statement, “Most GTs consistently show superior performance over GNNs across all datasets” (line 451), would be more convincing if compared with recent GNN baselines, such as ChebNetII[2] and OptBasis[3].
>
> R2. Thank you for pointing out the imprecise description.  The correct description would be: "Most GTs consistently show superior performance over **the backbone** GNNs, which typically are GCN and GAT, across all datasets.” Based on your advice, we further compare the proposed DUALFormer with these two recent GNN baselines, namely ChebNetII and OptBasis, on five datasets. As can be seen from the following table on five datasets, the proposed DUALFormer consistently outperforms the baseline GNNs. This underscores the effectiveness of DUALFormer.
>
> |  | Roman-Empire | Question  | ogbn-papers100M | pokec| ogbn-arxiv
> |:--------:|:--------:| :---------:|:--------:| :--------:| :--------:|
> | ChebNetII | 74.64$_{±0.39}$ |   74.41$_{±0.58}$ | 67.18$_{±0.32}$ | 82.33$_{± 0.28}$| 72.32$_{±0.23}$|
> | OptBasisGNN | 76.91$_{±0.37}$ |   73.82$_{±0.83}$ | 67.22$_{±0.15}$ | 82.83$_{±0.04}$ | 72.27$_{± 0.15}$|
> | DUALFormer |  77.31$_{±0.17}$ | 78.62$_{±0.56}$ | 67.59$_{±0.28}$ | 82.97$_{±0.43}$| 73.71$_{±0.22}$|
>
> ---
> >Q3. There are a few typographical errors.
>
> R3. Thanks for your careful review. We will meticulously check the manuscript to ensure all errors are corrected.

---

> ### Author Response · Authors · 2024-11-15
> **Response to Reviewer CPJ4 (Part 2)**
>
> >Q4. The first question concerns the reasonableness of applying softmax to the global correlations between features. Could you clarify these differences and explain why it is reasonable to replace $\phi(\mathbf{Q})(\phi(\mathbf{K})^{\top}\mathbf{V})$ with $\mathbf{Q}\operatorname{softmax}(\mathbf{K}^{\top}\mathbf{V})$?
>
> R4. The introducted softmax is just a implementation strategy, while the obvious equivalence $\phi(\mathbf{Q})(\phi(\mathbf{K})^{\top}\mathbf{V}) = (\phi(\mathbf{Q})\phi(\mathbf{K})^{\top})\mathbf{V}$ is the key point we want to emphasize. This equivalence motivates the additional transformer on feature dimension and the proposed DUALFormer. To demonstrate the ignorability of softmax, we conduct an ablation study on the impact of softmax with results shown in the following tables. It illustrate that we can employ $\phi(\mathbf{Q})(\phi(\mathbf{K})^{\top}\mathbf{V}) $, which is equivalent to $(\phi(\mathbf{Q})\phi(\mathbf{K})^{\top})\mathbf{V}$. We sincerely apologize for any confusion caused by the introduced softmax and will remove it in the final version.
>
> |  | Cora | CiteSeer  | PubMed | Computers | Photo  | CS | Physics |
> |:--------:|:--------:| :---------:|:--------:| :--------:| :---------:|:--------:|:--------:|
> | without softmax | 85.69 | 74.55  | 83.62 | 93.29 | 96.91 |   95.61 |  97.30 |
> | with softmax |  85.88 | 74.45 | 83.97 | 93.16 | 96.74 | 95.62 | 97.42 |
>
> ---
> > Q5. The  interpretation of the proposed global attention on the special case of one-dimensional feature.
>
> R5.  For the case of a one-dimensional feature, **neither** previous approaches **nor** the proposed DUALFormer gather information. $\phi(\mathbf{Q})\phi(\mathbf{K})^T$ in previous methods reduces to a rank-1 matrix, whose rows only differ from each other by a factor since $\phi(\mathbf{K})^T$ is a row vector and $\phi(\mathbf{Q})$ is a column vector. Thus, the aggregation patterns/coefficients for different nodes, represented by the rows of $\phi(\mathbf{Q})\phi(\mathbf{K})^T$, only differ by this factor. As a result, $\phi(\mathbf{Q})\phi(\mathbf{K})^T\mathbf{V}$ degrades to the same aggregation pattern/coefficient for different nodes. Since the essence of aggregation is the different aggregation patterns/coefficients for different nodes, previous approaches lose this characteristic for a one-dimensional feature. Therefore, they also do **NOT** gather information as the proposed DUALFormer in this special case.
>
> ---
> > Q6. The motivation for the study is not fully convincing.
>
> R6. We hope the above two responses could clarify the rationality of our motivation.  Firstly, the key motivation is the obvious equivalence $\phi(\mathbf{Q})(\phi(\mathbf{K})^{\top}\mathbf{V}) = (\phi(\mathbf{Q})\phi(\mathbf{K})^{\top})\mathbf{V}$. Second,  **neither** previous approaches **nor** the proposed DUALFormer gather information for the case of a one-dimensional feature. Thanks for your special case. **It also demonstrates the importance of multiple features for the transformer.  Thus, the proposed DUALFormer is further justified by exploring the correlation among multiple features with an additional transformer.**

---

> > ### Comment · Reviewer_CPJ4 · 2024-11-22
> >
> > Thank you for your comprehensive response. Most of my concerns have been addressed, and I have accordingly increased my score.

---

### Official Review · Reviewer_YCph · 2024-11-02

**Soundness:** 3
**Presentation:** 4
**Contribution:** 3
**Rating:** 6
**Confidence:** 4

**Summary:**

This paper introduces DUALFormer, a novel Graph Transformer model designed to address scalability challenges and improve local-global information fusion. The approach is both simple and theoretically grounded. Extensive experiments demonstrate DUALFormer’s effectiveness, scalability, and robustness.

**Strengths:**

1. This paper is well-motivated.
2. The proposed method is simple and effective.
3. The inclusion of theoretical analysis strengthens the work.
4. Extensive experiments show the effectiveness, scalability and robustness.
5. This paper is easy to follow.

**Weaknesses:**

1. The proposed method can be interpreted as "attention on attributes". I wonder how is it different from the standard self attention. Especially why it can perform better on node classification? And when it is expected to perform better and when not?
2. Can you provide further analysis, such as case studies, to further explain the semantic meanings of the "attention on attributes"?
3. Can you provide further analysis and empirical studies to show that the GNNs after the graph Transform can indeed learn the localities in graphs?

I will raise my score if my concerns are properly addressed.

**Questions:**

N.A.

---

> ### Author Response · Authors · 2024-11-15
> **Response to Reviewer YCph**
>
> >Q1. The proposed method can be interpreted as "attention on attributes". I wonder how is it different from the standard self attention. Especially why it can perform better on node classification? And when it is expected to perform better and when not?
>
> R1. Firstly, both the proposed self-attention on dimension regarding attributes and standard self-attention are to capture global information despite their different forms. The former approximately describes the global dependence between nodes, which is the main role of the latter, by characterizing the correlation between features. Secondly, the performance boost is not due to this alone but rather the dual design of local and global modules. This prevents the trade-off between local and global information and enables comprehensive information modeling. Finally, the proposed method improves the performance of GNNs by capturing the relationships between features. They perform better when there is a strong correlation among features, and their effectiveness may be less effective when such correlations are weak.
>
> ---
> >Q2. Can you provide further analysis, such as case studies, to further explain the semantic meanings of the "attention on attributes"?
>
> R2. The semantic meaning of the proposed attribute (feature) attention is that it focuses on the correlation among features, allowing the model to capture the information that is most discriminative for the task. We would like to provide the following case to illustrate this point.
>
> Suppose there are five nodes with four features, where three of these nodes (the index are the first three) belong to one class, and the other two belong to the other class. When the feature matrix exhibits low-class discriminability, the matrix can be exemplified by $\mathbf{H}=$
>
> [[ $\frac{1}{3}$,  $\frac{2}{3}$, 0, 0 ],
>
> [ $\frac{2}{3}$, $\frac{1}{3}$, 0 , 0 ],
>
> [ 0, 1, 0, 0 ],
>
> [ 0, 0, 1, 0 ],
>
> [ 0, 0, 0, 1 ]]$_{5\times 4},$ where the rows correspond to nodes and columns to features.
>
> Assuming a clear feature correlation, e.g., the first two features signal the first class, while the last two features correspond to the second class. The attention score matrix can be expressed as $\mathbf{S}=$
>
> [[ $\frac{1}{2}$,  $\frac{1}{2}$, 0, 0 ],
>
> [ $\frac{1}{2}$, $\frac{1}{2}$, 0 , 0 ],
>
> [ 0, 0, $\frac{1}{2}$, $\frac{1}{2}$ ],
>
> [ 0, 0, $\frac{1}{2}$, $\frac{1}{2}$ ]]$_{4 \times 4}$.
>
> Using feature attention, the updated features can be expressed as $\hat{\mathbf{H}}= \mathbf{H}\mathbf{S} =$
>
> [[ $\frac{1}{2}$,  $\frac{1}{2}$, 0, 0 ],
>
> [ $\frac{1}{2}$, $\frac{1}{2}$, 0 , 0 ],
>
> [ $\frac{1}{2}$, $\frac{1}{2}$, 0, 0 ],
>
> [ 0, 0, $\frac{1}{2}$, $\frac{1}{2}$ ],
>
> [ 0, 0, $\frac{1}{2}$, $\frac{1}{2}$ ]]$_{5\times 4}.$
>
> The updated features exhibit more obvious class discriminability compared to the input features.
>
> ---
> >Q3. Can you provide further analysis and empirical studies to show that the GNNs after the graph Transform can indeed learn the localities in graphs?
>
> R3. We would like to provide the following theoretical analysis to explain that the GNN module is able to learn the locality of the graph.
>
> Firstly, from the perspective of graph learning, many classical GNNs (e.g., GCN and SGC) can be induced by optimizing the objective function [1, 2], namely
>
> $tr(\mathbf{H}^{\top}\tilde{\mathbf{L}}\mathbf{H})=\frac{1}{2}\sum_{v,u}\tilde{a}_{v,u}\Vert \mathbf{h}_v-\mathbf{h}_u\Vert_2^2$,
>
> where $\tilde{\mathbf{L}}$ denotes the Laplacian matrix of the normalized adjacent matrix $\tilde{\mathbf{A}}$, and $\mathbf{H}$ stands for the node features such as $\mathbf{H}=\mathbf{X}\mathbf{W}$ in GCN and $\mathbf{H}= \mathbf{X}$ in SGC. This indicates that GNNs essentially learn local information through feature updates that are constrained by the graph topology.
>
> From the above perspective, the GNN module in the proposed DUALFormer is equivalent to solving the above objective function with $\mathbf{H}=\mathbf{Z}$, that is, $tr(\mathbf{H}^{\top}\tilde{\mathbf{L}}\mathbf{H})$, where $\mathbf{Z}$ denotes the node features obtained from the self-attention on the dimension regarding features. Thus, even as a post-processing technique, the GNN module can ensure localizing property by leveraging graph topology to constrain the feature updates.
>
> [1] Interpreting and Unifying Graph Neural Networks with An Optimization Framework. WWW 2021
>
> [2] Why Do Attributes Propagate in Graph Convolutional Neural Networks? AAAI 2021

---

> > ### Comment · Reviewer_YCph · 2024-11-17
> >
> > I would like to thank the authors for their rebuttal. However, I am confused by the paper and the authors’ rebuttal, especially on weakness 3. According to the paper, the graph Transformer can learn the global representation and the GNN model behind it can learn some local representation. However, the output of the GT (global representation) is the input to the GNN. Why the GNN can still learn local representation given global inputs?

---

> > > ### Author Response · Authors · 2024-11-20
> > > **Response to Reviewer YCph**
> > >
> > > >Q1. Why the GNN can still learn local representation given global inputs?
> > >
> > > R1. We appreciate your feedback and understand your concerns about the local expressivity of the proposed model. We would like to offer a more nuanced explanation, focusing on two key aspects.
> > >
> > > From a macro perspective, the features obtained from the attention module can be seen as information-rich representations, which the GNN then refines through local message passing to capture the locality of graphs. This aligns with the optimization perspective of graph learning mentioned in the last responses.
> > >
> > > From a micro perspective, these obtained features can be split into two representations: the self-representation and the globally aggregated representation. Accordingly, the GNN module in the proposed model serves as a shared component that updates these representations and eventually merges the updated features. To provide an intuitive understanding, we would like to present the following example.
> > >
> > > Let $\mathbf{H}$ stands for the initial node representation. Note that the global attention module is designed to aggregate all related features based on the attention score matrix. It is evident that the diagonal elements of the attention matrix are non-zero, as each entity is inherently related to itself.
> > >
> > > Then, according to the diagonal and off-diagonal elements of the attention matrix, the aggregated representation can be decomposed into two parts: the self-representation $\mathbf{H}\mathbf{W}$ (with coefficients corresponding to the diagonal elements) and the global aggregated representation $\tilde{\mathbf{H}}\mathbf{W}$ (with coefficients corresponding to the off-diagonal elements). Omitting the parameter $\mathbf{W}$, the updated representation can be formulated as $\hat{\mathbf{H}}=\mathbf{H}+\tilde{\mathbf{H}}$
> > >
> > > As a result, the outputted representations of the GNN module can be formulated as $GNN(\hat{\mathbf{H}})=GNN(\mathbf{H})+GNN(\tilde{\mathbf{H}})$. Given that $GNN(\mathbf{H})$ is an obvious local representation, it can be concluded that the final representation incorporates local information.

---

> > > > ### Author Response · Authors · 2024-11-27
> > > >
> > > > Thanks for your insightful feedback; it has significantly enhanced the quality of our paper. We have carefully answered your concerns and made the necessary revisions to the manuscript. Please let us know if you have any further questions. We are more than willing to provide explanations or clarification to ensure a thorough understanding of our paper.

---

### Meta-Review · Area_Chair_C352 · 2024-12-18

**Metareview:**

In this submission, the authors proposed a new member of Graph-oriented Transformer models with advantages in performance and computational efficiency. In particular, the authors proposed a new architecture separating local and global self-attention modules, in which a linearized Transformer is applied to reduce the complexity of the global self-attention module. AC and reviewers agree that the study of Graph Transformer is an important topic for the community, and the design of the proposed model is reasonable to some extent, making the model applicable for large-scale graphs.

In the rebuttal phase, the authors provided detailed feedback, including more analytic experiments and explanations. At the same time, the paper was revised carefully. The reviewers' concerns, which are mainly about the rationality of the architecture and the solidness of the experiments, have been resolved successfully.

In summary, AC decided to accept this work.

**Additional Comments On Reviewer Discussion:**

The reviewers interacted with the authors. Most of the reviewers were satisfied with the authors' rebuttals and increased their scores. After reading the submissions, the comments, and the rebuttals, AC has decided to accept this work.

---

### Decision · Program_Chairs · 2025-01-22

Accept (Poster)